# Differing pre-industrial cooling trends between tree-rings and lower-resolution temperature proxies

Lara Klippel[1], Scott St. George[2], Ulf Büntgen[3,4,5,6], Paul J. Krusic[3,7,8], Jan Esper[1]

[1]Department of Geography, Johannes Gutenberg University, Mainz, Germany

[2]Department of Geography, Environment and Society, University of Minnesota, Minneapolis, Minnesota, USA

[3]Department of Geography, University of Cambridge, Cambridge, United Kingdom

[4]Swiss Federal Research Institute for Forest, Snow, and Landscape (WSL), Birmensdorf, Switzerland

[5]Global Change Research Institute of the Czech Academy of Sciences (CzechGlobe), 603 00 Brno, Czech Republic

[6]Department of Geography, Faculty of Science, Masaryk University, 613 00 Brno, Czech Republic

[7]Department of Physical Geography, Stockholm University, Stockholm, Sweden

[8]Navarino Environmental Observatory, Messinia, Greece

*Correspondence to*: Lara Klippel (L.Klippel@geo.uni-mainz.de)

**Abstract.** The new PAGES2k global compilation of temperature-sensitive proxies offers an unprecedented opportunity to study regional to global trends associated with orbitally-driven changes in solar irradiance over the past two millennia. Here, we analyse pre-industrial long-term trends from 1 to 1800 CE across the PAGES2k dataset and find that, in contrast to the gradual cooling apparent in ice core, marine and lake sediment data, tree rings do not exhibit the same decline. To understand why tree-ring proxies lack any evidence of a significant pre-industrial cooling, we divide those data by location (high NH latitudes vs. mid latitudes), seasonal response (annual vs. summer), detrending method, and temperature sensitivity (high vs. low). We conclude that the ability of tree-ring proxies to detect pre-industrial, millennial-long cooling is not affected by latitude, seasonal sensitivity, or detrending method. Caution is advised when using multi-proxy approaches to reconstruct long-term temperature changes over the entire Common Era.

# 1 Introduction

Apart from documentary archives (Pfister et al., 1999), our estimate of climate variability prior to the systematic collection of instrumental measurements in the mid-nineteenth century relies on climate-sensitive proxy data (Christiansen and Ljungqvist et al.,2017; Frank et al., 2010; Jones et al., 2009; Smerdon and Pollack, 2016). Paleotemperature information can be extracted from natural archives such as ice cores (Steig et al., 2013), speleothems (Martín-Chivelet et al., 2011), tree-rings (Esper et al., 2014), lake and marine sediments (Nieto-Moreno et al., 2013), and glacier fluctuations (Solomina et al., 2016),

among others (Jones et al., 2009; Wanner et al., 2008). Today there are a number of multiproxy (Christiansen and Ljungqvist, 2012; Hakim et al., 2016; Hegerl et al., 2007; Jones et al., 1998; Ljungqvist et al., 2012; Mann et al., 2008; Mann et al., 2009; Neukom et al., 2019; Pages 2k Consortium, 2013; Pages 2k Consortium, 2019; Shi et al., 2013), and tree-ring only reconstructions (Briffa, 2000; D'Arrigo et al., 2006; Esper et al., 2002; Schneider et al., 2015; Stoffel et al., 2015; Wilson et al., 2016) of Northern Hemisphere (NH) and global temperatures. These reconstructions offer defensible

characterizations of pre-instrumental, naturally forced climate variability at annual resolution and millennial timescales (Christiansen and Ljungqvist, 2017; Wanner et al., 2008; Wanner et al., 2015), which is essential for placing Anthropogenic warming in a long-term context. Proxy data themselves provide valuable climate information needed to test and verify paleoclimate model simulations (Braconnot et al., 2012; Fernández-Donado et al., 2013; Hartl-Meier et al., 2017; Ljungqvist et al., 2019,  PAGES Hydro 2k Consortium, 2017;  PAGES 2k- PMIP3 group, 2015).


The PAGES2k database is a product of a community effort organized by PAGES (http://pastglobalchanges.org), to amass the world's largest collection of proxy records covering the Common Era (CE) (PAGES2k Consortium, 2017). The PAGES2k database 2.0.0 contains 692 temperature-sensitive proxy records from: trees (415), ice cores (49), lake (42) and marine sediments (58), corals (96), documentary evidence (15), sclerosponges (8), speleothems (4), boreholes (3), bivalves

(1), and a hybrid tree/borehole (1) from 648 locations distributed across all continents and major oceans (Fig. 1 and Fig. S1). Unlike previously published multiproxy compilations (Mann et al., 2008; PAGES2k Consortium, 2013), the database includes substantially more evidence from sources other than tree-rings, and many more records that cover the first millennium, thereby expanding the spatial and temporal coverage over oceanic and polar regions (PAGES2k Consortium, 2017). The number, spatial distribution, and diversity of the dataset provides an unprecedented opportunity to analyse

regional to large-scale temperature patterns over the Common Era. The PAGES2k Consortium (2017) produced a collection of global mean composites from each of the major proxy types in its dataset. Here we present a similar visualization using only the PAGES2k NH records. The average NH composites of all proxies including marine sediments, lake sediments, and glacial ice cores (Fig. 2a) exhibit strong negative trends that are consistent with the gradual pre-1800 cooling reported previously in other major syntheses of Holocene proxies (cited previously). By contrast, the NH composite, derived solely

from tree-ring records (Fig. 2b), shows the rapid post-1800 increase but no trend from 1-1800 CE. A pre-industrial cooling can be attributed to gradual changes in orbital forcing, shown to be an important driver of Holocene long-term climate

oscillations (Milanković, 1941; Wanner et al., 2015). Changes in solar insolation (Huybers and Curry, 2006) are caused by variations in the Earth's tilt (obliquity), orbit (eccentricity) and rotation axis (precession). Over the Common Era, precession triggers a shift of the Perihelion (the closest point between sun and Earth) from December to January (Berger, 1978; Berger and Loutre, 1991). The collective effects of eccentricity, precession and obliquity reduces NH warm season (June-August) incoming solar radiation by ~9 W/m$^2$ at 90°N, 5.5 W/m$^2$ at 60°N, and 3.4 W/m$^2$ at 30°N, and increases Southern Hemisphere warm season (December-February) radiation by ~3.8 W/m$^2$ at 90°S, 4.1 W/m$^2$ at 60°S, and 5 W/m$^2$ at 30°S (Laskar et al., 2004) (Fig. 3). These long-term changes in orbital forcing should, theoretically, affect regional temperatures differentially (Masson-Delmotte et al., 2013).

The lack of a long-term negative trend in the average global tree-ring record could be related to the difficulty of retaining such low-frequency variance in dendrochronological timeseries (Cook et al., 1995). Esper et al. (2012) demonstrated that orbital trends are retained in a long and well-replicated maximum latewood density (MXD) chronology, whereas such variability could not be preserved in the tree-ring width (TRW) data of the same trees. Esper et al. (2012) argues that, unlike long MXD records, tree-ring width (TRW) records are incapable of capturing orbital trends. If this is the case, then including TRW records in past global temperature assessments might result in an underestimate of pre-instrumental warmth, e.g. during Medieval and Roman Times (Esper et al., 2004; Frank et al., 2010; Wang et al., 2014). Combining proxies that systematically vary in their low-frequency trends seemingly contributes to the development of temperature reconstructions of differing temperature amplitudes over the pre-industrial era (Christiansen and Ljungqvist, 2012; Christiansen and Ljungqvist, 2011; D'Arrigo et al., 2006; Hakim et al., 2016; Jones et al., 1998; Juckes et al., 2007; Ljungqvist et al., 2012; Mann et al., 1999; Mann et al., 2008; PAGES2k Consortium, 2013; Schneider et al., 2015; Steiger et al., 2018; Wilson et al., 2016). Here we analyse the PAGES2k collection of temperature-sensitive proxy records to understand why the mean tree-ring record lacks a pre-industrial millennial-scale cooling trend that is otherwise preserved in ice core, lake and marine sediment data. We hypothesize that the absence of this long-term negative trend in tree-ring chronologies may be a consequence of the climate sensitivity of the trees used, their detrending, and spatial distribution of the datasets. To test these potential explanations, we explore the effect of three significant attributes of just the tree-ring component that may have bearing on the long-term temperature trend reported in the PAGES2k initiative.

(1) Based on the spatial and seasonally varying effect of orbital forcing over the Common Era, we expect a millennial-scale cooling trend prior to the industrial period, particularly in summer-sensitive, high northern latitude proxies (Esper et al., 2012; Kaufman et al., 2009). Therefore, the absence of a distinct pre-industrial cooling in the PAGES2k tree-ring network could be a by-product of the spatial distribution of tree-ring proxies in the network. If the 2k network had equal representation from mid- and high-latitude tree-ring records, it should capture the millennial-length cooling trend in summer,

as we expect proxy records from high northern latitudes to contain a stronger summer cooling trend than their mid-latitude counterparts.

(2) All tree-ring parameters, with the possible exception of $\delta^{18}O$ (Esper et al., 2015; Helama et al., 2015; Young et al., 2011), include age-related, non-climatic signals that need to be removed prior to chronology development and reconstruction (Bräker, 1981; Cook, 1990; Douglass, 1919; Fritts, 1976). The selection of a suitable tree-ring detrending method is one of the fundamental challenges in the field of dendroclimatology (Briffa et al., 1992; Cook et al., 1995; Esper et al., 2004; Melvin et al., 2013). However, tree-ring detrending methods vary in their approach to model tree growth and if applied indiscriminately can remove long-term cooling trends related to orbital forcing, either intentionally or inadvertently, interpreted as biological noise (Cook et al., 1995; Esper et al., 2004). Given that the PAGES2k database contains no information regarding the detrending method used to produce the tree-ring chronologies in its collection, we assume all were produced using different detrending methods, and that those methods are applied to differently structured tree-ring datasets (i.e. the temporal distributions of short and long tree-ring measurements series, indicative of young and old trees, over the past 2k years are not the same). If this is the case, such disparities will affect the database chronologies' long-term variability, causing the tree-ring mean to lack millennial scale trends (Briffa et al., 2013; Büntgen et al., 2017; Linderholm et al., 2014).

(3) The inclusion of chronologies having a mixed climate sensitivity (e.g. Seim et al., 2012) and their potential introduction of non-temperature related noise (Baltensweiler et al., 2008) might weaken a reconstruction. The establishment of large-scale (continental or hemispheric) temperature reconstructions relies on the assumption that all proxy records used to produce the reconstruction have a substantial temperature signal, and that the signal is temporally stable over the entire record length (Esper et al., 2016). We assume the inclusion of tree-ring chronologies with a mixed sensitivity, including other climate parameters besides temperature (Babst et al., 2019; Babst et al., 2013; Galván et al., 2014; Klesse et al., 2018), weakens a reconstruction, and that reconstructions composed of weakly calibrating chronologies contain less or no orbitally forced trends.

We begin by describing the varying ability of the proxies used in the PAGES2k network to preserve orbitally forced, millennial-scale, temperature trends. Then we evaluate and discuss how a more discriminating proxy selection might help improve our understanding of past climate variability over the Common Era.

## 2 Data and methods

### 2.1 Data preparation

The PAGES2k database (Fig. 1) was accessed via the website of the NCEI-Paleo/World Data Service for Paleoclimatology (https://www.ncdc.noaa.gov/paleo/study/21171). The Southern Hemisphere was excluded from our analysis due to having

too few samples (111 records in total, with only 13 tree-ring records) and the suggestion of ambiguous links between the hemispheres on orbital timescales (Kawamura et al., 2007; Laepple et al., 2011; Petit et al., 1999). All NH records were normalized over their individual record lengths by subtracting the time series mean ($\mu$) from each single proxy value, then dividing the difference by the series' standard deviation ($\sigma$). Normalization is a necessary step to eliminate differences in measuring scale, as the database includes a variety of measured parameters, including $\delta^{18}O$ (Horiuchi et al., 2008), TRW (Luckman and Wilson, 2005), MXD (Klippel et al., 2018), blue intensity (Björklund et al., 2014), varve thickness (Moore et al., 2001) or Sr/Ca (Rosenheim, 2005). We appreciate that the choice of normalization period, from which we calculate $\mu$ and $\sigma$, has an influence on the expression of long-term trends as seen in the tree-ring data (Fig. 4). Using $\mu$ and $\sigma$ of all the tree-ring chronologies' common period (1758-1972) leads to a slightly different millennial-scale trend compared to the PAGES2k procedure of using the individual records' total lengths. Large trend discrepancies arise from using $\mu$ and $\sigma$ of even shorter periods (e.g., 1800-50, 1850-1900 and 1900-50; Fig. 4). A $\mu_{sub\ period}$ and $\sigma_{sub\ period}$ smaller, or a $\mu_{sub\ period}$ and $\sigma_{sub\ period}$ larger, than the entire time series $\mu$ and $\sigma$ produces records with increased or decreased temperature levels and trends, respectively (Fig. S2). By normalizing all the proxies in the same manner, we minimized the influence of the normalization method on the preservation of long-term variability in tree-rings.

All proxy records having a negative correlation with instrumental temperature were inverted (multiplied by -1) to ensure that high proxy values represent warm temperatures and low proxy value cold temperatures. This procedure was applied to one marine sediment and four lake sediment records. To account for the varying temporal resolution among the proxies, from sub-annual to multi-decadal scale, all normalized records were averaged and set to the same resolution consisting of 50-year bins (e.g. 1901-1950; 1951-2000; Fig. 4). To test the influence of bin size on low-frequency variability, the normalized proxy records were also degraded to the 200-year resolution (Fig. S3). Test results show that bin size has no influence on the strength of the pre-industrial trend.

We realize that the normalization and binning procedure influences the strength of the pre-industrial trend and low-frequency variability. Reversing the order of binning and normalization produced an increase in low-frequency variability. Discrepancies between glacier ice, marine and lake sediment composite chronologies, and the tree-ring composite remain unchanged (Fig. S4). We propose the use of binned and scaled chronologies, because potential biases due to changing resolutions, e.g. sub-annual to multidecadal, are mitigated. However, since we need to conform with the procedures established by the PAGES2k Consortium (2017) we used their normalization and binning approach.

**2.2 Hypothesis testing**

To test the influence of (i) orbital forcing, (ii) tree-ring detrending and (iii) temperature sensitivity, we extracted a subset of proxy records from the PAGES2k database, restricted to only those records longer than 800 years. This 800-year threshold is

based on the reasonable assumption that longer records are more likely to express stronger millennial-scale trends. The subset includes 89 tree-ring, 16 glacier ice, 44 marine and 29 lake sediment records.

55

(1) Based on the Milankovitch cycles (Milanković, 1941) we expect latitudinally and seasonally varying temperature trends, with the strongest cooling to be found in summer-sensitive proxies from high-latitude, and the least cooling to be found in the annual temperature sensitive proxies from lower latitudes (Berger and Loutre, 1991; Laskar et al., 2004). To assess the long-term trends preserved in an individual tree-ring record from the PAGES2k compilation (which does not report the specific detrending method used for each entry), the statistical significance of the slopes from least-squares linear regressions through each proxy record (at 50-year and 200-year resolution) were evaluated, and the fraction of records that exhibited a significant or insignificant cooling trend over the pre-industrial period (1-1800 CE), and a warming trend over the industrial (post 1800 CE) period were recorded. For those tree-ring records that do not span the entire pre-industrial period, the slope calculation was performed over their entire length. Those records with significant warming and cooling trends were further analysed with respect to proxy type (archive), latitude, and temperature sensitive seasonality. These analyses were repeated over the proxies' common period 1200-1800 CE as well as with only those records that span the entire Common Era. The latter constrains the network to only 11 tree-ring, 10 glacier ice, 8 marine sediment and 6 lake sediment records. To account for the bias due to an inhomogeneous distribution of sites along a latitudinal gradient, we randomly selected 1000 times ten records from latitudinal bands 0-90°N, 30-60°N and 60-90°N to determine the number of records showing an (in-) significant cooling/ warming over the pre-industrial period. In addition, we produced 50-year and 200-year binned records (tree composite versus glacier ice, marine sediment and lake sediment composite) for each latitudinal band, to illustrate chronology trend changes along the gradient. Additionally, we explored the influence of the absolute record length on the strength of the pre-industrial cooling.

75

(2) As noted previously, the PAGES2k compilation does not include mention of the detrending method used to produce each tree-ring chronology. To address this omission, we re-standardized the tree-ring records, to test how the choice of detrending method used affects the resulting chronologies' millennial scale trend. Of the 89 chronologies selected, the raw data of 22 datasets could not be obtained from either the International Tree-ring Databank (ITRDB) or the original authors. Consequently, this aspect of our analysis focuses on a subset of 67 chronologies. The tree-ring detrending methods applied are the calculation of residuals from individually fit (i) negative exponential functions (NEG), and (ii) from regional growth curves (RCS; Briffa et al., 1992; Esper et al., 2003). The individual series detrending method (i) emphasizes annual to centennial trends in the resulting index chronology (Cook and Peters, 1981) by removing long-term trends that exceed the lengths of sampled trees. By contrast, RCS (ii) attempts to preserve low-frequency climate variability through its address of the so called "segment length curse" (Briffa and Melvin, 2011; Cook et al., 1995). However, traditional RCS is best applied to large datasets with a homogenous age-structure through time to optimise the ideal representation of the population growth curve used to detrend the data (Esper et al., 2003), and most tree-ring measurements in the 2k database do not satisfy this

criterion. To address this trend distortion due to increasing tree age over time, we applied a third detrending method (iii) Signal-Free Regional Curve Standardization was performed (RCS-SIG; Melvin et al., 2014). Prior to detrending, a data adaptive power transformation was applied to all measurements to mitigate the heteroscedastic nature of the tree-ring series (Cook and Peters, 1997), and chronologies calculated using the bi-weight robust means of tree-ring indices in each calendar year. In addition, the average correlation coefficient among the individual series (Rbar; Wigley et al., 1984) was used to stabilize the variance of the chronologies (Frank et al., 2007). The resulting chronologies from each of the three methodologies i, ii, and iii were then z-transformed and averaged over 50-year bins to produce three unique composite chronologies. The 50-year binned composites were compared with the PAGES2k subset composite that includes the same 67 records to investigate the influence of tree-ring standardization on millennial scale temperature trends.

(3) The nature of the climate signal encoded in each tree-ring record was assessed by Pearson correlation coefficients between all 402 NH z-transformed tree-ring chronologies, the subset of 89 NH tree-ring chronologies, and both the 1° and 5° gridded CRU TS 4.01 (Harris et al., 2014) monthly June-September temperatures from 1950-1980. The relatively short interval of 31 years was selected for computing correlations in response to the sparse station data availability, especially in Asia, and the decline in the quality of interpolated observational temperature data prior to 1950 (Cook et al., 2012, Krusic et al., 2015). For each re-standardized and z-transformed chronology, the highest monthly maximum correlation coefficient was extracted and plotted with respect to the trees' location as provided in the metadata table (PAGES2k Consortium, 2017). The use of extended calibration periods (prior to 1950 and post 1980), and annual temperatures, yielded no meaningful differences in the calibration results. The stability of the growth-climate relationship was assessed by first smoothing the tree-ring and corresponding CRU temperatures using 10-year splines then using the splines to high-pass filter the data and accentuate inter-annual variances. The tree-ring records were ranked according to the strength of their maximum monthly temperature response between June and September, and averaged into 50-year binned composites to evaluate the importance of changing signal strength of any preserved millennial-scale trend.

## 3 Results

### 3.1 Latitude and season

In total, 66.3% of the tree-ring, 93.8% of the glacier ice, 75.0% of the marine and 79.3% of the lake sediment records, longer than 800 years, reveal a millennial-scale cooling over the period 1-1800 CE (Fig. 5a). Substantial differences between the proxies were apparent when comparing the fraction of records with a significant overall cooling trend ($p < 0.05$): 68.8% of the glacier ice, 54.5% of the marine and 37.9% of the lake sediment records, but only 11.2% of the tree-ring records. Sorting the data by latitude reveals that the fraction of significantly cooling tree-ring records decreases from 25.0% at 60-90°N to 8.7% at 30-60°N, which, though the percentages are fairly small, supports the argument that the signature of orbital forcing

in tree-rings has a meridionally declining spatial signature. In contrast, the cooling trends in glacier ice, marine and lake sediment records reach their maximum in the mid-latitudes, from 30-60°N, which contradicts this explanation. This finding remains stable even after repeating our analysis by 1000 times, each time randomly drawing 10 unique combinations of tree-rings records or composites of glacier ice, marine and lake sediments. Pre-industrial cooling remains significantly stronger in glacier ice, marine and lake sediment records compared to tree-ring records at different latitudinal bands. This result indicates clearly that differing pre-industrial cooling trends are not by-product of the spatially varying distribution along a latitudinal gradient (Fig. S5). Organizing the chronologies with respect to latitude confirms that glacier ice, marine and lake sediment records from the mid-latitudes (30-60°N) show an enhanced cooling compared to their high-latitude counterparts (60-90°N), whereas the NH tree-ring composites lack any significant cooling (Fig. S6). The overall number of summer temperature sensitive proxy records showing long-term cooling is similar to the number of annual temperature sensitive proxies showing long-term cooling, suggesting that the orbitally forced reduction in summer insolation over the past 2k years has no substantial effect on the expression of long-term trends. Considering only the common period 1200-1800CE, to investigate pre-industrial trends, leads to a substantially different result. Only 7.8% of the tree-ring proxies show a significant pre-industrial cooling, as opposed to 18.8% of the glacier ice, 15.4% of the marine sediment and 19% of the lake sediment records, suggesting potential trend issues related to the absolute length of the chronologies (Fig. 5b). However, there exists no clear relationship between the strength of this trend and the absolute record length (Fig. S7). As an additional test, pre-industrial cooling trends where analysed in records that span the entire Common Era (Fig. 5c). Use of the very longest records (1-1800CE), again reveals substantial proxy differences. A significant pre-industrial cooling appears in 9.1% of the tree-ring, 80% of the glacier ice, 75% of the marine sediment and 33% of the lake sediment records. Over the industrial period, 1800-2000 CE, glacier ice, marine and lake sediments, and tree-ring records particularly, consistently show a temperature increase (Fig. 5e).

**3.2 Tree-ring detrending**

We applied three different detrending methods with varying ability to preserve low-frequency information on a subset of 67 of the 415 datasets in the PAGES2k database. The single best replicated collection is the Torneträsk (Sweden) TRW dataset containing 650 measurement series. The least replicated is a dataset from southern China containing just 10 measurement series. This huge range of underlying data points to potential weaknesses in our application of RCS, which requires high sample replication so common climate-driven variability does not affect the estimate of the regional growth curve (Briffa et al., 1992, Esper et al., 2003). Comparisons between our NEG, RCS, RCS-SIG composite and the PAGES2k subset composites, reveals how there is substantially more low-frequency variability present in the RCS and RCS-SIG chronologies (Fig. 6). Extended cool periods are from 500-750 CE, 1450-1500 CE and 1600-1800 CE, and prolonged warm periods between 850-1200 CE and 1800-2000 CE. Cooling is more pronounced in the RCS chronology compared to the RCS-SIG

chronology, whereas the latter has an increased industrial-era warming. In the NEG and PAGES2k subset composite, pre-industrial temperature variations are restricted to multi-decadal scales, indicating cool conditions from 250-300 CE and 1450-1500 CE, warm conditions from 550-600 CE, and a more persistent warming from 1850 CE to present. Comparison of the RCS and RCS-SIG detrended composites against the PAGES2k tree-ring composite reveals substantial differences in long-term trends in the first millennium. This demonstrable difference is a consequence of the pronounced cooling from 500-750 CE, a feature lacking in the both the PAGES2k subset (Fig. 6) as well as entire PAGES2k tree-ring composite (Fig. 2), but conserved in the RCS and RCS-SIG mean chronologies. Good agreement exists in the second millennium, as the magnitude, timing and strength of warm and cool intervals largely overlap. The best fit over the entire Common Era exists among the NEG and PAGES2k subset composites, suggesting the PAGES2k database includes a sizable amount of NEG detrended records. However, even with the application of RCS, arguably the best current method for preserving long-term trends in tree-rings when suitably applied, the pre-industrial cooling trend in the PAGES2k tree-ring dataset differs significantly from those found in glacier ice, marine and lake sediment records (Fig. 2 and Fig. 6).

### 3.3 Climate signal strength

Pearson correlation analyses between the tree-ring proxy records and their respective local temperature grids reveals considerable inter-continental differences in the proxy's response to maximum monthly June-September temperature (Fig. 7a). The median correlation coefficients differ substantially by region, reaching 0.6 in the Arctic (contributed by the PAGES2k Arctic regional), 0.21 in Asia, 0.54 in Europe and 0.38 in North America. Associations between temperature and tree growth are higher in the Arctic (87.5% of records are significant correlated with maximum June-September temperatures) and Europe (75%), but the agreement between proxies and climate observations are weaker in Asia (21%) and North America (61%). However, these differences might be an artefact of different sampling strategies. In the first case (Arctic and Europe), only 16 and 8 highly temperature sensitive records are considered, but Asia and North America have 228 and 150 records respectively. The differences among the continents, as demonstrated by the distributions of their June-September correlation coefficients, remains fairly stable in the different frequency domains, as well as for records longer than 800 years (Fig. 7b). To account for seasonal responses beyond the June-September window, and potential influences of the calibration period, the analysis was repeated for all months, varying warm season means, and extended calibration periods (1950 to the end dates of the individual chronologies). No substantial changes were recorded (not shown). Despite significant differences in high-to-low-frequency temperature signals, we find that none of the composites, integrating the good, medium and poorly calibrating records, contain a significant millennial-scale cooling (Fig. 7c-d). This result suggests climate signal strength is not related to the long-term trends present in tree-ring chronologies.

## 4 Discussion

### 4.1 Orbital signatures in regional and large-scale records

Our results show that millennial-scale trends in NH proxy records are consistent between tree-ring, glacier ice and marine and lake sediment records when considering the period 1200-1800 CE to calculate the slope of pre-industrial trends but inconsistent between tree-rings and other proxies over the entire Common Era. Despite a non-systematic relationship between record length and the slope of a pre-industrial trend, this finding demonstrates the majority of proxy records that only cover large parts of the second millennium, fail to preserve a significant negative long-term trends over the entire Common Era. In contrast to glacier ice and marine and lake sediment records, most of the very longest tree-ring records covering the entire pre-industrial Common Era 1-1800 CE do not exhibit a long-term cooling. The high-latitude tree-ring based temperature reconstruction from Scandinavia remains the only record with a significant pre-industrial cooling (Esper et al., 2012).

The signature of orbital forcing has been described in regional studies from the Arctic and Antarctica (Esper et al., 2012; Kaufman et al., 2009; Kawamura et al., 2007), as well as in one Holocene climate reconstruction based on a multiproxy collection from the northern high- and mid-latitudes; the latter attributing a distinct value to the orbital cooling effect of 0.5°C since the Holocene Thermal Maximum (Marcott et al., 2013; Routson et al., 2019). However, in the case of Marcott et al. (2013), it has been shown that NH cooling is only apparent in high-latitude North Atlantic proxies, and that the trend would not exist without them (Marsicek et al., 2018). Previous studies have also reported that it is difficult to reconcile the negative orbital forcing trends preserved in proxy data with simulated temperatures which show a strong warming of about 0.5°C over the Holocene (Liu et al., 2014, Laepple et al., 2011). From a theoretical perspective, independent of the proxy type, we would expect a stronger cooling trend in summer temperature proxies and an increase in the strength of the trend from the mid to the high-latitudes (e.g., Esper et al., 2012; Kaufman et al., 2009). The absence of a clear meridional and seasonal pattern demonstrates the importance of internal climate variability (Deser et al., 2010; Schneider and Kinter, 1994) and other external forcing factors (Sigl et al., 2015; Vieira et al., 2011) on proxy records. We conclude that although multiple tree-ring datasets are systematically limited in their low-frequency amplitude, they deviate from forcing expectations in the same way as all other proxies. We conclude that the reduced low-frequency variability in tree-ring data cannot be explained by an overrepresentation of the mid-latitudes in the hemispheric composite.

Despite the insignificant pre-industrial temperature changes in 86.5% of the tree-ring records, compared to other proxies, the post 1800 CE warming trend in tree-rings is significant (25.8% versus 11.9%). Consequently, large scale multiproxy climate reconstructions that include long tree-ring records (> 800 years), or solely tree-ring based reconstructions developed from the PAGES2k database, will likely show a stronger post-1800 warming than multiproxy reconstructions that choose to exclude

(long) tree-ring records (Fig. 2 and Fig. 6). The selection of the proxy type has major implications on the reconstructed warmest interval over the Common Era. Using marine data, the warmest period is 151-200 CE and the pre-industrial Era is dominated by a strong cooling trend, suggesting the magnitude of the current warming is not outstanding. By contrast, in lake sediments, ice cores, and tree-ring data, the most recent period is exceptionally warm (Fig. 2). This finding highlights the importance of tree-ring data in any effort to determine whether, over the past two millennia, the twentieth-century and early twenty-first century temperatures are unprecedented in both their magnitude and rate of warming (Büntgen et al., 2011; Foley et al., 2013).

## 4.2 The impact of detrending on temperature trends

The degree of similarity between the NEG tree-ring chronology produced here and the corresponding PAGES2k version suggests that the current PAGES2k tree-ring collection is not the most ideal for studying millennial scale trends. This is in large part due to the limitations of individual series detrending (Cook et al., 1995). Even with the application of RCS and RCS-SIG (Briffa et al., 1992; Esper et al., 2003, Melvin et al., 2014), the detection of a millennial-scale cooling trend is still elusive. These findings clearly demonstrate that the limited low-frequency variance in tree-ring chronologies is not solely an artefact of individual series detrending, previously identified as main explanation for the observed lack of long-term oscillations in large scale temperature reconstructions (Esper et al., 2002). Our reassessment of tree-ring chronologies also highlights the importance of the detrending methodology in reconstructing centennial scale temperature variability, as evidenced by the performance of the RCS and RCS-SIG chronologies. In both we can clearly identify the Late Antique Little Ice Age (LALIA) (Büntgen et al., 2016), a cool period from 300-750 CE that is absent in the PAGES2k version, albeit with slightly greater uncertainty about the mean. The Büntgen et al. (2016) analysis and the dataset used in this study only share four tree-ring records in common, thus our analysis provides independent confirmation of the existence of LALIA and cooler conditions during the Migration period (Büntgen et al., 2011). In contrast, during LALIA the PAGES2k tree-ring time series suggest a period of alternating warm and cool decades, but no persistent cooling on large spatial scales.

## 4.3 Temperature sensitivity and the link to long-term trends

Temperature sensitivity was a key criterion for inclusion into the PAGES2k database (PAGES2k Consortium, 2017) and was assessed by the PAGES community through comparison with gridded HadCRUT 4.2 temperatures (Morice et al., 2012). Our analysis has shown the PAGES2k database includes many tree-ring records that have a weak relationship with local temperature at high-to-low frequencies. The monthly maximum correlation coefficients between 1x1° CRU TS 4.01, June-September temperature data falls below 0.2 in 126 cases. The lowest correlation coefficient is -0.25 (unfiltered data). Such week temperature sensitivities amongst the tree-rings is likely related to confounding non-climatic (Johnson et al., 2010; Konter et al., 2015) or hydroclimatic (Ljungqvist et al., 2016) growth controls, or to the circumstance that some records are

by nature less sensitive to summer temperature than others (St. George, 2014). Further contributions to the extreme range of PAGES2k tree-ring proxies', climate signal strength is related to the fact that MXD chronologies more sensitive to temperature than TRW chronologies (Büntgen et al., 2009). At the same time, some records might be more temperature sensitive than they appear due to their calibration against noisy or inappropriate temperature targets (Böhm et al., 2009; Cook et al., 2012). The re-calibration against instrumental temperatures showed that temperature sensitivity and absolute climate signal strength are of limited importance for the preservation of millennium scale cooling trends in tree-ring records. Even the best calibrated records (r > 0.6; 1950-1980) convey a different low-frequency signature compared to the glacier ice, marine, and lake sediment records. This observation is relevant to the current debate in paleoclimatology on optimal strategies for compiling proxy datasets to represent past natural temperature variability: is it best to include (a) a large number of proxy records, including those possessing a relatively weak temperature signal, or (b) a small number of only the very best calibrated proxies (Christiansen and Ljungqvist, 2017).

## 4.4 Remaining uncertainties

This work examines the influence of orbital forcing, tree-ring detrending and climate signal strength on pre-industrial cooling in marine and lake sediment, glacier ice and tree-ring proxy archives. In tree-ring chronologies, sample size decreases back in time, lowering the chronology's signal-to-noise ratio and increasing variance (Frank et al., 2007). A small sample size could create apparent trends in the composite chronology that are not real. Regrettably, critical information about the sample replication for each tree-ring chronology is not completely provided by the PAGES 2k initiative (PAGES2k Consortium, 2017) and thus we speculate records were truncated according to community-wide standards. Furthermore, the influence of climate epochs during the Common Era; the Roman Optimum (Büntgen et al., 2011); the Medieval Warm Period and Little Ice Age (Grove, 1990) on pre-industrial temperature trends has not yet been systematically explored. The magnitude, timing and duration of the warming and cooling during these phases spatially varies and it has been shown that there is globally no spatiotemporal coherence between cold and warm epochs exists (Neukom et al., 2019). Further exercise potentially requires assessment of the relationship between timing and magnitude of climate epochs and overall temperature trends (Frank et al., 2010).

## 5 Conclusion

The community-sourced database of 692 different temperature-sensitive proxy records in the PAGES2k initiative provides unprecedented opportunities to study long-term temperature trends at regional to global scales. Combining glacier ice, marine and lake sediment records that span the Common Era reveals a persistent, millennial-scale cooling over the pre-

industrial period that is missing in tree-ring data. Our analysis has shown that the observed discrepancies in long-term trends do not arise from the latitudinal and seasonally varying imprints of orbital forcing or the limited temperature sensitivity. Despite application of the most suitable tree-ring detrending, one that can potentially support the preservation of low-frequency temperature trends at millennial time scales, substantial long-term trend differences between proxies remain. We conclude that some, possibly many of the tree-ring records in the PAGES2k database are artificially limited in their low-frequency variance at centennial and longer time scales due to individual series detrending, This observation is supported by the fact that when a more low-frequency conserving tree-ring detrending method is applied to a large subset of suitable records, new corroborating evidence for the existence of the LALIA appears. Such nuances in the affect various tree-ring detrending methods have on low-frequency variance conservation needs to be considered when combining proxies in large scale temperature reconstructions to avoid the underrepresenting late Holocene cooling trends prior to post-industrial warming in hemispheric and global mean temperature reconstructions.

**Data availability.**

The PAGES2k database was accessed via the website of NCEI-Paleo/World Data Service for Paleoclimatology (https://www.ncdc.noaa.gov/paleo/study/21171).

**Author contributions.**

JE and SSG were the leaders of the project. PK and UB contributed to the planning and structuring of the analysis and publication. LK performed the analysis and wrote the manuscript with contributions from all co-authors.

**Competing interests.**

The authors declare that they have no conflict of interest.

**Acknowledgements.**

This research was supported by the German Science Foundation, grants # Inst 247/665-1 FUGG and ES 161/9-1, and the Alexander von Humboldt Foundation. We thank Alexander Kirdyanov, Hans Linderholm, Fredrick C. Ljungqvist, Alma Piermattei, Denis Scholz, and Eduardo Zorita for discussion and helpful comments. Three referees, Ed Cook, Lea Schneider and one anonymous referee made thoughtful and detailed comments that improved the quality of the manuscript.

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

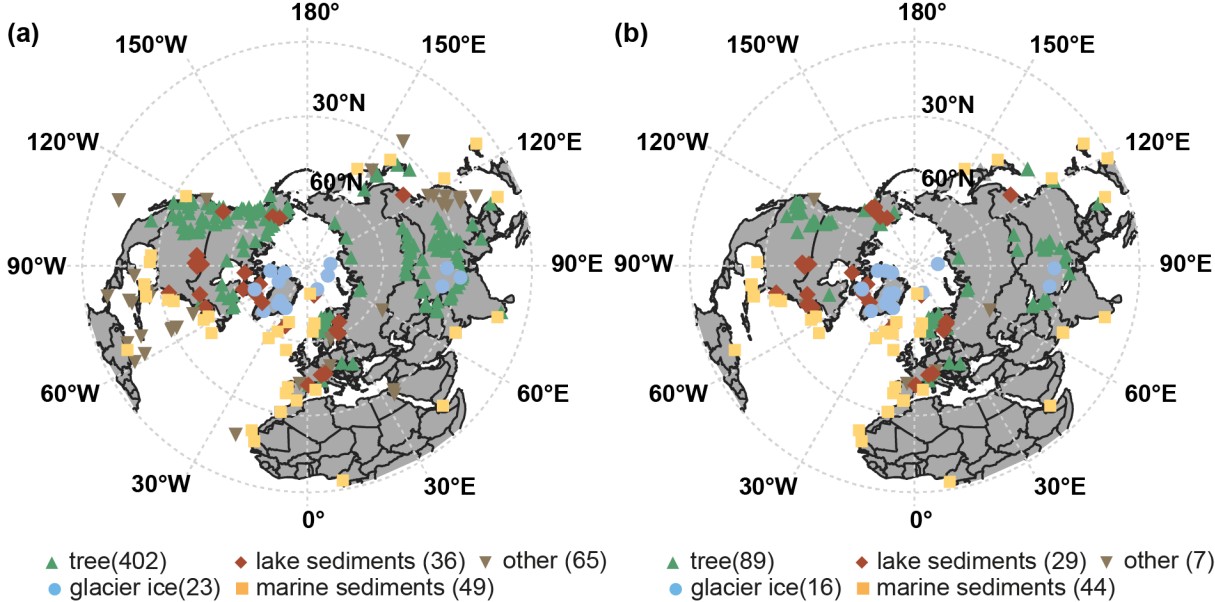

**Fig. 1. (a)** Map showing the spatial distribution of Northern Hemisphere proxy records from the PAGES2k 2.0.0 database including primary tree-ring (green), glacier ice (blue), marine (orange) and lake (red) sediment records as well as a smaller number of records from bivalves, boreholes, corals, documents, hybrids, sclerosponges, and speleothems (brown). **(b)** same as (a) but showing only those records longer than 800 years.

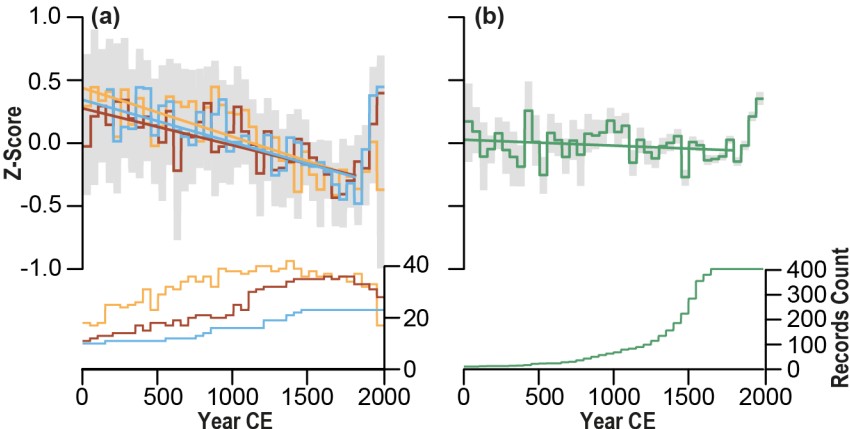

**Fig. 2.** Compilation of NH temperature-sensitive proxy records from the PAGES2k initiative. **(a)** 50-year binned composites from 49 marine sediment (orange), 36 lake sediment (red) and 23 glacier ice (blue) records expressed in standard deviation units. Straight lines highlight the pre-industrial temperature trends (1-1800 CE) and lower panels show the corresponding temporal distribution of the records. Grey shadings indicate 95% bootstrap confidence intervals with 500 replicates. **(b)** same as in (a) for 402 tree-ring records.

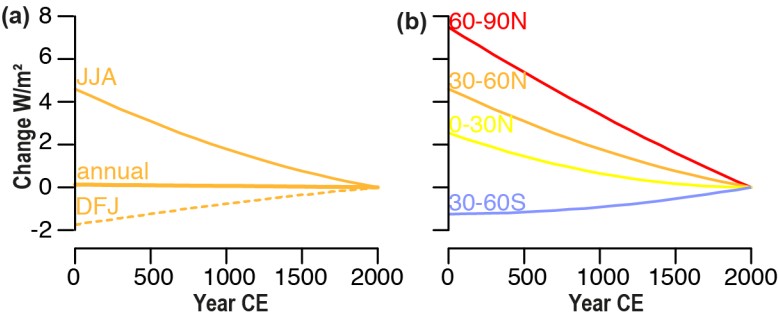

**Fig. 3. (a)** June-August, December-February, and annual insolation changes at 30-60°N relative to 2000 CE and **(b)** June-August insolation changes at different latitudinal bands (Laskar et al., 2004).

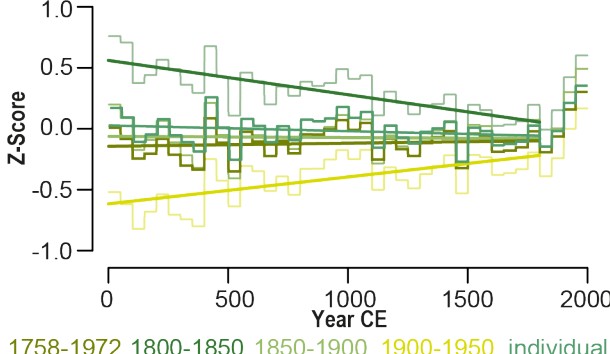

**Fig. 4.** Effect of tree-ring normalization on long-term temperature trends. NH composite tree-ring records from 402 records normalized using the means and standard deviations over different time spans.

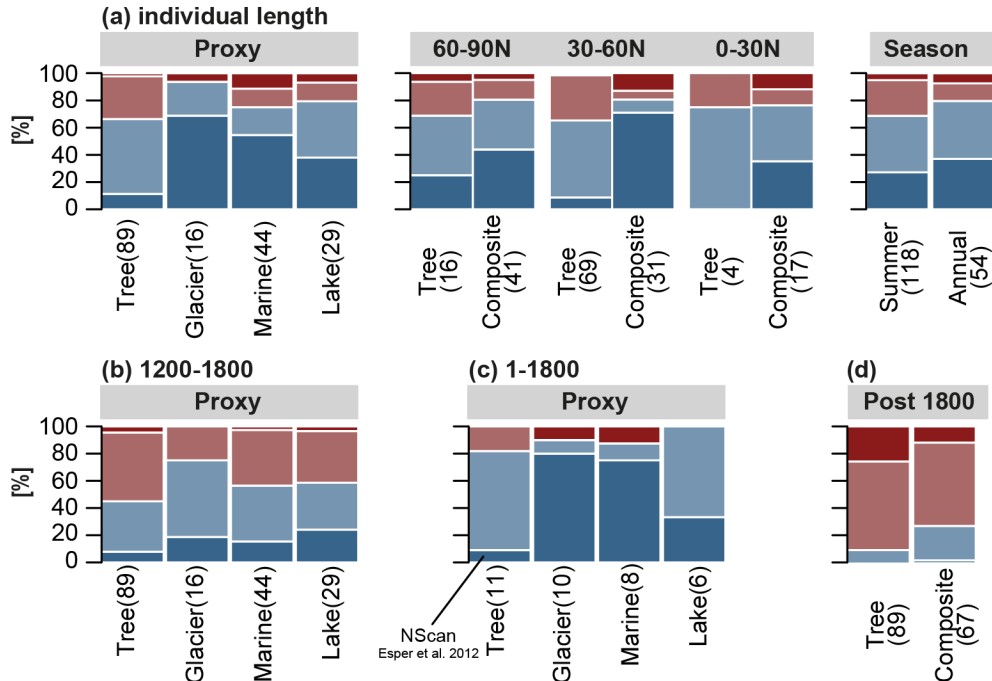

**Fig. 5.** Summary of NH long-term trends from tree-ring, glacier ice, marine and lake sediment records longer than 800 years. The fraction of 50-year binned records that exhibit a significant negative (dark blue) and non-significant cooling (blueish) trend or significant (red) and non-significant (reddish) warming trend at $p < 0.05$ over the pre-industrial (1-1800) period derived from the statistical significance of the slope of least-squares linear regressions through each individual 50-year binned proxy record using **(a)** the individual records length, **(b)** the common period 1200-1800, and **(c)** records that cover the entire Common Era. Pre-industrial summaries are split by proxy, latitude, and seasonality. The category composite includes glacier, marine and lake sediments, and brackets indicate the number of records per category. **(d)** Post-industrial trends over the period 1800-2000.

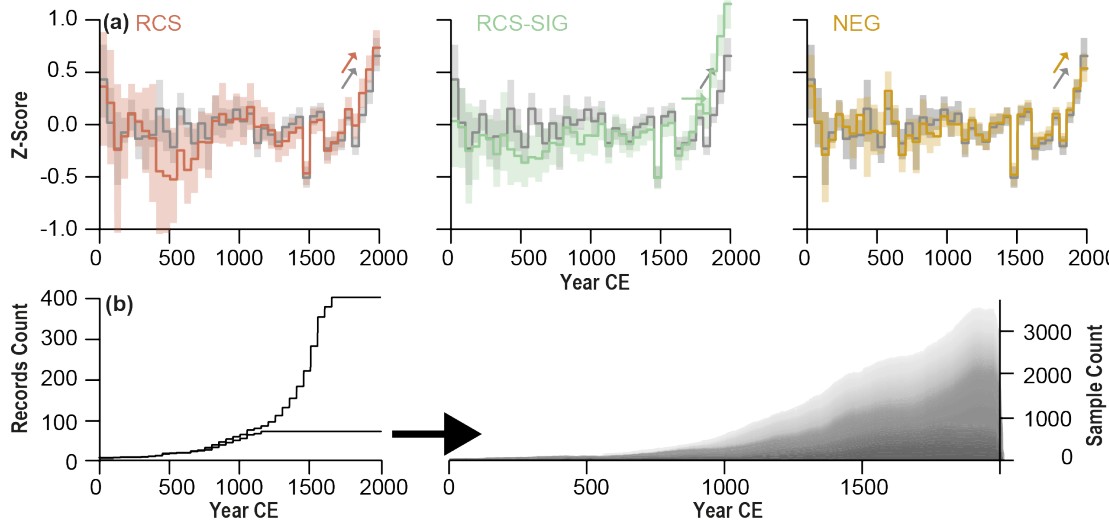

**Fig. 6.** Effects of tree-ring detrending on long-term trends. **(a)** 50-year binned composites from 67 RCS (red), RCS-SIG (green) and NEG (gold) standardized datasets. The PAGES2k composite (dark grey) includes the corresponding chronology versions that are provided in the 2.0.0 database. Shadings indicate 95% bootstrap confidence intervals with 500 replicates, and the arrows indicate the direction of the post-1800 trend. **(b)** Temporal distribution of the NH tree-ring samples (402) relative to the detrended subset (67) and distribution of individual samples from records included in the subset (grey shadings).

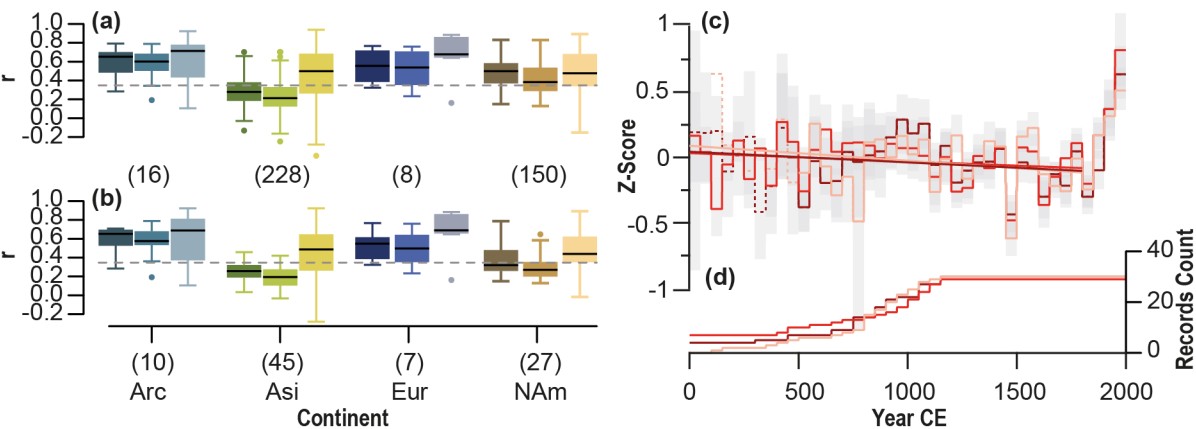

**Fig. 7.** Effects of tree-ring calibration on long-term temperature trends. **(a)** Maximum correlation coefficients between NH individual site-level tree-ring records from the Arctic (Arc), Asia (Asi), Europe (Eur) and North America (Nam) and 1x1° CRU TS 4.01 June-September monthly temperature data over the period 1950-1980, divided by region, using 10-year high-pass filtered data (left box), original data (central box), 10-year smoothed data (right box). Dashed line indicates the $p < 0.05$ threshold. **(b)** Same as (a) using only records longer than 800 years, and corresponding **(c)** 50-year binned composites divided by climate signal strength including records with the lowest (n= 30; rose), medium (n= 31; red) and highest (n= 30; dark red) climate sensitivity. Light grey shadings indicate 95% bootstrap confidence intervals with 500 replicates and **(d)** temporal distribution of the records.