# Peer review of "Differing pre-industrial cooling trends between tree-rings and lower-resolution temperature proxies"

_Climate of the Past, 2019_

## Referee Comment (RC1) · Edward Cook (Referee) · 25 Apr 2019

Review of "Differing pre-industrial cooling trends between tree-rings and lower-resolution temperature proxies" by Lara Klippel, Scott St. George, Ulf Büntgen, Paul J. Krusic, Jan Esper

Submitted by: Edward R. Cook

This is a useful paper that illustrates in part some important short-comings of the tree-ring data available from the PAGES 2k database 2.0.0 for the detection of millennial-long temperature trends in total ring widths (TRW). The motivation for this evaluation

was based on the observation that other temperature-sensitive proxies in the PAGES 2k database show what is interpreted as an orbitally-driven pre-industrial cooling trend over the past 2,000, whereas temperature-sensitive tree-ring data based on ring widths in the same database do not do so. This cooling trend should be found in the Northern Hemisphere (NH) as theory predicts; in the Southern Hemisphere (SH) an opposite warming trend should be found, again based on theory. These expected hemispheric insolation trends are also most strongly expressed in the boreal and austral summers, respectively, with amplification towards the poles (Fig. 3). This should favor the more northerly tree-ring series in the NH because they are principally summer temperature responders, yet a cooling trend is not apparent. Although it is briefly mentioned in the Abstract that the seasonal response could also be 'annual' (line 18), it is practically impossible to convincingly argue that tree-ring series anywhere reflect changes in mean annual temperature in their total ring widths. This persistent mythology needs to be laid to rest.

The authors note that there are 415 "temperature-sensitive" tree-ring chronologies in the global PAGES 2k database and proceed to use them to illustrate the lack of a pre-industrial cooling trend in tree-ring series relative to the other temperature proxies (Fig. 2). The 50-year binned composite tree-ring series do have a very slight negative trend, but it is not statistically significant compared to the much larger negative trends found in the other proxies. While this initial comparison sets the stage for the investigations carried out in the rest of the paper, it is somewhat strange because by my calculation (using the 402 NH chronologies noted in Fig. 1a) there are 415-402=13 SH tree-ring series in the 415-chronology total that could have orbitally-driven positive temperature trends in them. Thus, including the 13 SH chronologies in the binned composite (Fig. 2b) likely weakens the chance of finding a statistically significant negative trend in the 402 NH chronologies. Figure 2b should therefore be redone using only the 402 NH chronologies for the binned evaluation of the temperature trend. I do not necessarily expect a change of outcome, but it should be done to be consistent with the argument. On line 60, the reference to "average global tree-ring record" should consequently be

changed to "average Northern Hemisphere tree-ring record" as well. Later in the paper it is finally mentioned that the SH tree-ring data were excluded from further analysis (lines 128-129). This should have been mentioned in the beginning, and the SH chronologies immediately excluded, as suggested above.

The lack of a significant negative trend in the tree-ring data (assuming to hold after the SH chronologies are removed) is hypothesized to come from three issues: climate sensitivity, detrending, and spatial distribution. Reasonable arguments for each being a contributor are given and each is tested. The issue of detrending is perhaps the one most dear to my heart because I have spent much of my career studying it. It was therefore with considerable dismay that "the PAGES 2k database contains no information regarding the detrending method used to produce the tree-ring chronologies in its collection . . ." (lines 90-91). Admittedly, the ITRDB holdings are not that much more informative. Even so, this lack of metadata is unfortunate and should be rectified as a matter of PAGES 2k policy. The lack of useful metadata on how the tree-ring series were detrended is why I almost never use tree-ring chronologies directly from the ITRDB. I detrend the raw measurements myself. The majority of my remaining comments will deal mostly with the detrending tests done in the paper.

At the hypothesis testing stage (pg. 5) the original 415 global chronologies were winnowed to a subset of 70 NH tree-ring collections, each at least 800 years long (line 143). It would be useful to have a table in the paper (as an appendix?) that lists these data sets by location, species, length, and modeled climate sensitivity (e.g., maximum correlation with monthly summer temperature), so that others can download the same tree-ring data and repeat the same evaluations on their own. I note that this number (70) is less than that indicated in Fig. 1b (89). Another apparent inconsistency. Regardless, selecting only the longer series for further evaluation is quite sensible because the preservation and detection of a negative pre-industrial temperature trend (if there) will be far easier to accomplish with $\geq$800 year-long series due to issues related to the 'segment length curse'.

This being the case, I cannot fathom why experiments with (i) 100-year spline detrending (SPL) were carried out. The result is totally predictable with respect to the preservation of multi-centennial to millennial-long trends; they are all removed. Therefore, no negative pre-industrial temperature trend can ever be expected to be detected. Thus, SPL detrending has no relevance to testing the effects of detrending on the presence or absence of the expected negative temperature trend. The use of (ii) negative exponential functions (NEG) gets closer to the issue of preserving multi-centennial to millennial-long variability, so it is useful to experiment with. However, its susceptibility to the 'segment length curse' renders it almost impossible for NEG to preserve a millennial-long cooling trend unless the series being detrended are least millennial in length. This leaves (iii) regional curve standardization (RCS) as the only detrending option that may preserve the negative pre-industrial temperature trend from ring widths being sought. Thus, the authors are right in stating that that this method is the best of the three. Unfortunately, they are also right in stating the tree-ring measurements in the PAGES 2k database are for the most part inadequate for the application of RCS detrending. Besides the datasets not being nearly large enough in general, there are other reasons detailed in Briffa and Melvin (2011) on why use of RCS on inappropriate data sets can lead to the creation of utterly spurious long-term trends. This is because it is rarely appropriate to use RCS on datasets based only on living trees from the same site. Yet, I suspect that this is the case for most of the datasets in the PAGES 2k database. Applying RCS to such datasets can introduce what Briffa and Melvin (2011) call 'modern sample bias' in the form of an artificial positive slope to the final RCS chronology, which is exactly the opposite to what is expected here based on orbital forcing. This being said, the 50-year binned RCS composite may still be useful to evaluate because any 'modern sample bias' in individual chronologies may be attenuated in the large-scale multi-site composite. This is still not optimal, however. On a positive note, I actually find the results shown in Fig. 5 to be encouraging. The 2/3 of the tree-ring chronologies (based on RCS? unclear in the text.) have negative trends up to 1800 CE. Although most are declared not statistically significant, it

would be useful to test for the probability that the combined outcome is in fact statistically significant in favor of a cooling trend being there more than by chance alone in 2/3 of the series. This can be easily done using Fisher's combined probability test (https://en.wikipedia.org/wiki/Fisher%27s_method) assuming (quite reasonably here) independence between the individual test outcomes. However, again there appears to be a mismatch in the number of series longer than 800 years declared for use (70) (line 143) and the number indicated in the left-hand bar chart in Fig. 5 (89). The same problem is apparent in the total chronology count (89) shown in Fig. 7b. This inconsistency must be corrected.

The results of the other two tested hypotheses – climate sensitivity and spatial distribution – appear reasonable to me, although it is unclear which version of the chronologies is being used; SPL, NEG, or RCS. This ambiguity must be corrected. I would, however, caution about using chronologies with "mixed climate sensitivity", i.e. a combination of temperature and moisture sensitivity in the ring widths. Published work by Matt Salzer and Andy Bunn show that even at the highest, most temperature limited, elevations in the White Mountains of California it is easy to find bristlecone pine trees that have mixed precipitation/temperature signals in their ring widths. When this occurs, the correlation with summer temperature can be negative, leading to the suggestion here to invert those chronologies to rectify the correlation with temperature (lines 125-126). I am not sure this is a good idea. This form of negative temperature sensitivity is completely different from that for trees with positive temperature sensitivity because the negative correlation with temperature most likely reflects an evapotranspiration demand signal associated with a positive response to soil moisture content and precipitation amount. It is not clear that one should expect this relationship to have the same trend (in inverted form) as that due to the direct effect of summer temperatures on radial growth because this mixed-signal climate response is likely to include the direct effect of changing precipitation amount, which behaves more like a 'white noise' process compared to temperature. At the very least, one might expect the temperature trend expressed in the inverted tree-ring series to be reduced by the effects of a

precipitation signal on ring width.

I do not mean to be overly critical of this paper. It should be published after considering my suggestions and comments. It is certainly plausible that TRW cannot provide useful estimates of millennial-long, orbitally-driven, summer temperature changes over the pre-industrial Common Era as Jan Esper would likely argue. There may be biological limitations on ring width that limit both the preservation of such long-term temperature-driven trends and their separability from purely biological trends. If so, even RCS may fail to preserve millennial-long trends due to climate. However, it would be premature to conclude that this is true based on the experiments conducted here. They are very useful, but the PAGES 2k tree-ring database is not sufficient. For RCS detrending, most of the data sets used are inadequate for a variety of reasons. Regardless, the results for tree rings in Fig. 5 are in my opinion actually quite encouraging given the data being evaluated. For this reason, I remain guardedly optimistic that tree-ring chronologies based on TRW will be able to detect an orbitally-driven millennial-long cooling trend in the NH.

---

## Referee Comment (RC2) · Lea Schneider (Referee) · 16 May 2019

The manuscript "Differing pre-industrial cooling trends between tree-rings and lower-resolution temperature proxies" is a worthwhile contribution to the paleoclimate literature. It investigates a notable lack of millennial scale trend in a compilation of tree-ring chronologies from the recently published PAGES2k database (PAGES2k Consortium 2017). The lack of cooling trend – hypothesized to be a result of orbital forcing – contrasts information from other archives such as marine and lake sediments as well as glacial ice. The authors analyze three potential reasons for the absence of this trend in many of the tree-ring data: (1) a latitudinal or seasonal bias in the tree-ring network,

(2) an inappropriate detrending applied to many of the tree-ring chronologies and (3) the climate signal strength. The PAGES2k database is the most extensive collection of temperature sensitive proxy records up to date. As such, it is of high relevance for large scale paleoclimate studies, although the selection criteria applied by the PAGES community are somewhat controversial. Quality assessments of this product, beyond the tests reported by the consortium in the Scientific Data study, are topical and of high relevance for all secondary users. If the authors can address the comments listed below, this will be a manuscript suitable for publication in Climate of the Past.

The message of the paper is somewhat discouraging: The largest collection of temperature sensitive tree-ring records is unable to preserve millennial scale trends. However, I'm not sure if the main reason is the proxy type (TRW vs. MXD) as suggested early on (P3 L63-64). Much more relevant seems the selection strategy for proxy records in large scale compilations. This study shows that the PAGES approach (i.e. basically maximizing the number of records) is unable to account for limitations of single records and I fully agree that, therefore, this compilation should be used very carefully.

Major comments 1) Although the different tests applied by Klippel et al. are meaningful and reasonable, I would like to suggest one other experiment that might explain some of the offset in trends. The data preparation in this study follows the steps outlined in the PAGES2k network study. However, the last step described in the PAGES study, a scaling to temperature, is not applied (for some unknown reason, data were also not scaled in the corresponding PAGES figure). For the significance of long term trends, the scale is irrelevant and I'm not suggesting a scaling to temperature. More importantly, I want to point out that binning (or any other sort of low-pass filtering) needs to be followed by a scaling to either standard normal deviates or temperature, if the frequency spectra of the original data are very different. The latter is to be expected according to the title of this manuscript. The signal of low resolution records will be inflated compared to the low frequency tree-ring signal if scaling precedes binning. I expect the weak negative trend in the tree-ring compilation over the 1-1800CE period

to become less weak compared to trends in other archives (Fig. 2) if scaling to a common target follows binning (or low-pass filtering). This is a common procedure in multiproxy studies (e.g. Ljungqvist et al. 2016). These considerations should not alter the significance of trends. However, even binned tree-ring records might still have a less negative slope in the frequency space compared to records with an originally low temporal resolution. Marine sediment records with 200 years time steps, which fulfil the PAGES selection criteria, should have no (non-random) loading at frequencies around 50 years and therefore a steeper negative slope. Having a higher proportion of variability at multidecadal scale (compared to millennial scale) might penalize tree-ring records when assessing the significance of linear trends over almost 2 millennia. Whether this effect is relevant or not, could be tested, e.g., by binning with 200 years intervals. This might decrease the difference between tree-rings and other archives in Fig. 5. 2) The significance of trends might be even more affected by the variable length of tree-ring records. Is there a relationship between the length of the records and the significance of trends? It is reported that trends were calculated over the 1-1800CE period, but it is not clear how the authors dealt with records terminating before 1CE. Even if only records of >800 years are selected, the vast majority of them will not cover the entire 1-1800CE period. I assume the trends were then calculated over the remaining period, e.g. from 1000-1800CE. The authors need to specify in which way they considered that a shorter record (i.e. less degrees of freedom) likely reveals less significant millennial scale trends. 3) The authors are a bit ambiguous in their terminology when it comes to the appropriateness of detrending methods. Although they acknowledge that RCS detrending is best applied to datasets with certain characteristics (L52-54), they term individual detrending methods as inappropriate (L64+102). I agree that individual detrending methods are often inappropriate to preserve low frequency trends. However, depending on the age structure and the replication of the dataset, RCS can be likewise inappropriate. Some authors of tree-ring based climate reconstructions consider such shortcomings by stating that their record cannot capture millennial scale trends, an information that is usually ignored when incorporating data in larger scale compilations.

Multiproxy data collectors are not necessarily dendrochronologists. Thus, it is vital to be more specific when discussing these aspects to keep dendroclimatology credible.

Minor comments P3 L61-65 Differences between TRW and MXD data are not discussed in this manuscript. Without testing the hypothesis that MXD is better able to preserve millennial scale trends, I suggest to remove these sentences in order to prevent wrong expectations among readers. P3 L74 Inhomogeneous spatial distributions and mixed climate signals are not only problems for the tree-ring component! In fact, I would guess that the average climate signal is much stronger among tree-ring records compared to other archives. P7 L14 Please define Arctic. P8 L41-42 But the trend is not only significant in the global (or NH) mean. Fig. 5 shows that about half of the records exhibit a significant trend at local scale. P9 L70-72 Instead of presenting the number of overlapping tree-ring chronologies it would be more helpful to report a percentage (although this might be more difficult under a constantly changing number of records).

---

## Referee Comment (RC3) · Anonymous Referee #3 · 17 May 2019

General Remarks: It is quite possible that the differences the authors describe between dendro records and the composite proxies (I will refer to all of the other proxies considered in the manuscript as composite proxies in the remainder of this review) in the PAGES2K network are real. I am nevertheless not convinced and have what amounts to two principle and related concerns:

1. The insufficient investigation of spatial sampling biases on the results 2. Insufficient evaluation of the statistical significance between the differences described in the two populations

With regard to 1, the authors appropriately discuss the latitudinal gradient in the or-

bitally forced trends in temperature. They nevertheless do little to describe and investigate the latitudinal sampling biases in the two populations that they explore, namely the dendro and composite proxy populations. This sampling bias is obvious in Figure 1 and in the sample sizes listed in Figure 5. And yet figures like Figure 2 are presented with little caveat. Such a figure is misleading, given that the composite records are biased toward the high latitudes and the dendro records are biased toward the midlatitudes and the lower midlatitudes in particular (incidentally, it is not mentioned anywhere whether these means are themselves weighted by cos(lat), as they should be). How should we interpret these time series given that the explored effect intensifies at the higher latitudes? Splitting Figure 2 into time series representing different latitude bands would help (30 degree boxes may be too large for this), as would a scatter plot of trends vs. latitude for each of the two populations. While not definitive, it would be helpful for understanding how spatial sampling of a spatially-dependent temperature trend may be biasing the mean trends estimated from the two populations. I suspect that the authors will bring up Figure 5 as a rebuttal to what I am pointing out, but please see my comments on my second principle concern below. Before I get to that, however, I would add that another overlooked bias is that of the proxies comprising the composite records. They all sample different seasonal windows, some reflect marine temperature changes as opposed to continental changes, and many have their own biases tied to representation of low-frequencies. The authors take the composite proxies at face value, presumably because they fit their assumptions about latitudinal trends (in most cases), but it is insufficient to do so. There may be biases in these other records that promote spurious trend estimates that the authors do nothing to highlight. One observation that may point to such biases is the increase in the percentage of significant mid-latitude trends in the composite records relative to the high latitude records, which is of course counter to the expected spatial dependence. These factors are not sufficiently discussed.

Regarding concern 2, the authors present Figure 5 as a measure of the latitudinal differences in the significance testing of trends in the dendro and composite records.

The percentage of each population with significant and insignificant trends is nevertheless hard to interpret. Some additional significance testing would go a long way toward helping to interpret this figure and the results. The first question that should be addressed is: given the expected magnitude of trends estimated a priori from the orbitally-forced changes in insolation (signal), time series with the level of variance representative of the proxy series (noise), and the size of the sample populations, how many times would one achieve significant positive/negative trends and insignificant positive/negative trends for different realizations of noise? For instance, it may be the case that for 16 time series and the level of variance that is estimated in each, the trend percentages in the dendro high-latitude bin is actually what you would expect for a modestly detectable trend. Moreover, how should we interpret the comparison between the percentages associated with the dendro and composite series in a band like the high latitudes in Figure 5? It may be that the PDF of the percentage distributions spans the differences shown in Figure 5 and the results are fully consistent with each other. Put differently, for multiple noise realizations, how robust is the separation between the trend percentages in the dendro and composite records? It is impossible to answer these questions from what the authors have done and they should better characterize the statistical likelihood that the differences they describe are more than just noise.

All of the above is fundamental for two reasons. The first is that the difference in the number of statistically significant trends is the primary metric by which the authors argue there are differences in the representations of trends between the two populations. If the physical expression of those trends is latitudinally dependent and their spatial sampling is heterogenous and biased in the two populations, it must be controlled for. Secondly, the robustness of the differences must also be statistically constrained so that real differences (statistically speaking) can be separated from differences that can arise simply by chance.

Minor Comments:

Note that my pdf copy appears to have the first digit of the line numbers cut off. Please interpret the line numbers below with that in mind.

Pg. 1, Ln. 13: It should be noted here that the 692 proxies are the temperature sensitive records in the database (the full database is closer to 3000).

Pg. 1, Ln. 24: There are a lot more reviews that speak to this issue than Frank et al. Consider adding Jones et al. (2009), North et al. (2006), Mann (2007), Smerdon and Pollack (2016), and Christiansen and Ljungqvist (2017).

Pg. 2, Ln 28: The list of multiproxy reconstructions does not include the data assimilation work (e.g. Hakim et al. (2016) and Steiger et al. (2018)) nor does it include the PAGES products from 2013 and 2018. This should be corrected.

Pg. 2, Ln 32: The list of review articles that discuss this should be expanded as above.

Pg. 2, Ln 34: This is once again a limited list of papers that compare reconstructions and models. The authors should at least include the PAGES efforts from PAGES2k-PMIP Group (2015) and PAGES Hydro2K Consortium (2017), if not include some of the additional references that are discussed in those studies.

Pg. 4, Ln 6: Consider discriminating instead of critical

Pg. 5, Ln 46: It seems strange to use cubic smoothing splines for standarization in the context of this investigation, given that they will explicitly remove the long-term trends. The effect is clearly visible in Figure 6 where even the 20th century trends have been removed. Incidentally, I find the bracket and description in Figure 6b a bit clumsy and hard to follow. The bracket in particular looks like it was drawn in by hand!

Pg. 6, Ln. 84-86: Doesn't this contradict a central premise of the paper? This seems a lot more concerning than the attention it is given in the manuscript.

Pg. 7, Ln 90: The subset is described as 70 but there are multiple places where this number appears to be different. Figure 5, for instance, discusses 89 dendro series.

Are these typos or am I missing something?

Pg. 7, Ln. 5: How does -0.32 compare to -0.03?!

Pg. 7, Ln. 9: Consider preserving instead of conserving

Pg. 8, Ln. 37: I find the discussion starting here and extending to the end of the paragraph very confusing. It seems to be saying that the authors have demonstrated differences between proxies, but that there are no differences between proxies. With regard to the last sentence, I do not think the authors have demonstrated the lack of spatial sampling bias, based on the principle arguments I have provided above.

––––––––––––––––––––––––––

---

## Author Response (AR1)

**(1) comments from Referees, (2) author's response, (3) author's changes in manuscript**

**Point- by – Point response Ed Cook**

**Comment 1:**

(1) The authors note that there are 415 "temperature-sensitive" tree-ring chronologies in the global PAGES 2k database and proceed to use them to illustrate the lack of a pre- industrial cooling trend in tree-ring series relative to the other temperature proxies (Fig. 2). The 50-year binned composite tree-ring series do have a very slight negative trend, but it is not statistically significant compared to the much larger negative trends found in the other proxies. While this initial comparison sets the stage for the investigations carried out in the rest of the paper, it is somewhat strange because by my calculation (using the 402 NH chronologies noted in Fig. 1a) there are 415-402=13 SH tree-ring series in the 415-chronology total that could have orbitally-driven positive temperature trends in them. Thus, including the 13 SH chronologies in the binned composite (Fig. 2b) likely weakens the chance of finding a statistically significant negative trend in the 402 NH chronologies. Figure 2b should therefore be redone using only the 402 NH chronologies for the binned evaluation of the temperature trend. I do not necessarily expect a change of outcome, but it should be done to be consistent with the argument. On line 60, the reference to "average global tree-ring record" should consequently be changed to "average Northern Hemisphere tree-ring record" as well. Later in the paper it is finally mentioned that the SH tree-ring data were excluded from further analysis (lines 128-129). This should have been mentioned in the beginning, and the SH chronologies immediately excluded, as suggested above.

(2) We acknowledge the point that there might arise a trend distortion.

(3) Changed, but not only for tree-ring records. Fig. 2a and Fig. 4 were changed accordingly.

**Comment 2:**

(1) At the hypothesis testing stage (pg. 5) the original 415 global chronologies were winnowed to a subset of 70 NH tree-ring collections, each at least 800 years long (line 143). It would be useful to have a table in the paper (as an appendix?) that lists these data sets by location, species, length, and modeled climate sensitivity (e.g., maximum correlation with monthly summer temperature), so that others can download the same tree-ring data and repeat the same evaluations on their own.

(2) Due to additional consideration of Signal Free (see below), the number of records had to be reduced to 67 NH records, because the program failed to detrend 3 of the datasets.

35      (3) Metadata table (selection of information provided by PAGES) was added to the appendix.

**Comment 3:**

(1) Regardless, selecting only the longer series for further evaluation is quite sensible because the preservation and detection of a negative pre-industrial temperature trend (if there) will be far

40      easier to accomplish with ≥800 year-long series due to issues related to the 'segment length curse'. This being the case, I cannot fathom why experiments with (i) 100-year spline de-trending (SPL) were carried out. The result is totally predictable with respect to the preservation of multi-centennial to millennial-long trends; they are all removed. Therefore, no negative pre-industrial temperature trend can ever be expected to be detected. Thus, SPL

45      detrending has no relevance to testing the effects of detrending on the presence or absence of the expected negative temperature trend.

(2) The 100-year spline detrending was added to show how close the results are compared to to a method removing multi-centennial variability. Anyway, we understand that true insiders find the result predictable.

50      (3) The 100-year spline detrending was removed (and Signal Free added. See below.).

**Comment 4:**

(1) This leaves (iii) regional curve standardization (RCS) as the only detrending option that may preserve the negative pre-industrial temperature trend from ring widths being sought. Thus,

55      the authors are right in stating that that this method is the best of the three. Unfortunately, they are also right in stating the tree-ring measurements in the PAGES 2k database are for the most part inadequate for the application of RCS detrending. Besides the datasets not being nearly large enough in general, there are other reasons detailed in Briffa and Melvin (2011) on why use of RCS on inappropriate data sets can lead to the creation of utterly spurious long-term

60      trends. This is because it is rarely appropriate to use RCS on datasets based only on living trees from the same site. Yet, I suspect that this is the case for most of the datasets in the PAGES 2k database. Applying RCS to such datasets can introduce what Briffa and Melvin (2011) call 'modern sample bias' in the form of an artificial positive slope to the final RCS chronology, which is exactly the opposite to what is expected here based on orbital forcing.

65      This being said, the 50-year binned RCS composite may still be useful to evaluate because any 'modern sample bias' in individual chronologies may be attenuated in the large-scale multi-site composite. This is still not optimal, however. On a positive note, I actually find the

results shown in Fig. 5 to be encouraging. The 2/3 of the tree-ring chronologies (based on RCS? unclear in the text.) have negative trends up to 1800 CE.

(2) Sure not all datasets are fully appropriate for applying RCS.

(3) As an additional test, we performed Signal Free Regional Curve Standardization that should cope with some of the biased in the TRW chronologies.

**Comment 5:**

(1) Although most are declared not statistically significant, it would be useful to test for the probability that the combined outcome is in fact statistically significant in favor of a cooling trend being there more than by chance alone in 2/3 of the series. This can be easily done using Fisher's combined probability test (https://en.wikipedia.org/wiki/Fisher%27s_method) assuming (quite reasonably here) independence between the individual test outcomes.

(2) We have chosen another option in line with a comment of reviewer 3. The reviewer claimed that we do not account for a latitudinal sampling bias. This was addressed by adding uncertainty estimates retrieved from MonteCarlo based tests.

(3) Figure S.4 was added.

**Comment 6:**

(1) However, again there appears to be a mismatch in the number of series longer than 800 years declared for use (70) (line 143) and the number indicated in the left-hand bar chart in Fig. 5 (89). The same problem is apparent in the total chronology count (89) shown in Fig. 7b. This inconsistency must be corrected.

(2) We are simply not able to access all 89 raw datasets, a circumstance we have to accept (and point to in our paper).

(3) Further information was added to explain the varying numbers.

**Comment 7:**

(1) The results of the other two tested hypotheses – climate sensitivity and spatial distribution – appear reasonable to me, although it is unclear which version of the chronologies is being used; SPL, NEG, or RCS. This ambiguity must be corrected.

(3) Information was added.

**Comment 8:**

(1) I would, however, caution about using chronologies with "mixed climate sensitivity", i.e. a com- bination of temperature and moisture sensitivity in the ring widths. Published work by Matt Salzer and Andy Bunn show that even at the highest, most temperature lim- ited, elevations in the White Mountains of California it is easy to find bristlecone pine trees that have mixed precipitation/temperature signals in their ring widths. When this occurs, the correlation with summer temperature can be negative, leading to the suggestion here to invert those chronologies to rectify the correlation with temperature (lines 125-126). I am not sure this is a good idea. This form of negative temperature sensitivity is completely different from that for trees with positive temperature sensitivity because the negative correlation with temperature most likely reflects an evapotranspiration demand signal associated with a positive response to soil moisture content and precipitation amount. It is not clear that one should expect this relationship to have the same trend (in inverted form) as that due to the direct effect of summer temperatures on radial growth because this mixed-signal climate response is likely to include the direct effect of changing precipitation amount, which behaves more like a 'white noise' process compared to temperature. At the very least, one might expect the temperature trend expressed in the inverted tree-ring series to be reduced by the effects of a precipitation signal on ring width.

(2) The inversion of records affected only some non-tree ring records.

(3) Explanation about the records that were inverted was added to the methods section.

**Point- by – Point response Lea Schneider**

**Comment 1:**

(1) Although the different tests applied by Klippel et al. are meaningful and reasonable, I would like to suggest one other experiment that might explain some of the offset in trends. The data preparation in this study follows the steps outlined in the PAGES2k network study. However, the last step described in the PAGES study, a scaling to temperature, is not applied (for some unknown reason, data were also not scaled in the corresponding PAGES figure). For the significance of long term trends, the scale is irrelevant and I'm not suggesting a scaling to temperature.

(2) We did not scale the data to temperature, to produce Figure 2 (this publication) because in Figure 8 of the original publication (Pages 2k 2017) the data were also not scaled to temperature. See the original caption: "Figure 8. 50-year binned composites stratified by archive type, for all types comprising 5 or more series. Composites with fewer than 10

available series are shown by a dotted curve, while solid lines indicate more than 10 series. Shading indicates 95% bootstrap confidence intervals with 500 replicates. Gray bars indicate the number of records per bin. The composites are expressed in standard deviation units, not scaled to temperature".

(3) No changes made.

**Comment 2:**

(1) More importantly, I want to point out that binning (or any other sort of low-pass filtering) needs to be followed by a scaling to either standard normal deviates or temperature, if the frequency spectra of the original data are very different. The latter is to be expected according to the title of this manuscript. The signal of low resolution records will be inflated compared to the low frequency tree-ring signal if scaling precedes binning. I expect the weak negative trend in the tree-ring compilation over the 1 1800CE period to become less weak compared to trends in other archives (Fig. 2) if scaling to a common target follows binning (or low-pass filtering). This is a common procedure in multiproxy studies (e.g. Ljungqvist et al. 2016). These considerations should not alter the significance of trends. However, even binned tree-ring records might still have a less negative slope in the frequency space compared to records with an originally low temporal resolution.

(2) This was tested by switching the procedure: First binning, the scaling (blue = glacier ice, orange = marine sediment, red = lake sediment, green = tree-ring records). Here we show both a reproduction of Figure 2 (upper panel) and the result of the suggested, reversed processes of binning followed by scaling (lower panel).

[Figure]

Reversing the binning/scaling procedure increases multi-decadal to centennial scale variability. However, this is the case for all proxies, i.e. not only for the tree-ring data. The reduced pre-industrial cooling in the tree-ring data remains the same.

(3) No changes added to the manuscript.

**Comment 3:**

(1) Marine sediment records with 200 year time steps, which fulfil the PAGES selection criteria, should have no (non-random) loading at frequencies around 50 years and therefore a steeper negative slope. Having a higher proportion of variability at multidecadal scale (compared to millennial scale) might penalize tree-ring records when assessing the significance of linear trends over almost 2 millennia. Whether this effect is relevant or not, could be tested, e.g., by binning with 200 years intervals. This might decrease the difference between tree-rings and other archives in Fig. 5.

(2) The test slightly changes the differences, however, the major discrepancies remain the same (see below).

[Figure]

Fig. 5 after using 200-year bins.

(3) No changes added to the manuscript.

**Comment 4:**

(1) The significance of trends might be even more affected by the variable length of tree-ring records. Is there a relationship between the length of the records and the significance of trends? It is reported that trends were calculated over the 1-1800CE period, but it is not clear how the authors dealt with records terminating before 1CE. Even if only records of >800 years are selected, the vast majority of them will not cover the entire 1-1800CE period. I assume the trends were then calculated over the remaining period, e.g. from 1000-1800CE. The authors need to specify in which way they considered that a shorter record (i.e. less degrees of freedom) likely reveals less significant millennial scale trends.

(2) We are aware of the problem, thus analysis was constraint to records longer than 800 years.

(3) Further information and explanation was added to the manuscript (Fig. S.3).

**Comment 5:**

(1) The authors are a bit ambiguous in their terminology when it comes to the appropriateness of detrending methods. Although they acknowledge that RCS detrending is best applied to datasets with certain characteristics (L52-54), they term individual detrending methods as inappropriate (L64+102). I agree that individual detrending methods are often inappropriate to

preserve low frequency trends. However, depending on the age structure and the replication of the dataset, RCS can be likewise inappropriate. Some authors of tree-ring based climate reconstructions consider such shortcomings by stating that their record cannot capture millennial scale trends, an information that is usually ignored when incorporating data in larger scale compilations. Multiproxy data collectors are not necessarily dendrochronologists. Thus, it is vital to be more specific when discussing these aspects to keep dendroclimatology credible.

(3) This is very correct, we adopted the text accordingly and by including Signal Free Regional Curve Standardization.

**Minor comments**

**Comment 6:**

(1) P3 L61-65 Differences between TRW and MXD data are not discussed in this manuscript. Without testing the hypothesis that MXD is better able to preserve millennial scale trends, I suggest to remove these sentences in order to prevent wrong expectations among readers.

(2) Even though it is not tested in the publication, we consider this a very important finding which needs to be considered in the introduction.

(3) No Changes.

**Comment 7:**

(1) P3 L74 Inhomogeneous spatial distributions and mixed climate signals are not only problems for the tree-ring component! In fact, I would guess that the average climate signal is much stronger among tree-ring records compared to other archives.

(2) Yes this is likely true. However, we have no expertise in assessing the limitations and strength of other archives. Thus we focus only on tree-ring records to perform this analysis.

(3) No changes made.

**Comment 8:**

(1) P7 L14 Please define Arctic.

(3) Explanation was added.

**Comment 9:**

    (1) P8 L41-42 But the trend is not only significant in the global (or NH) mean. Fig. 5 shows that about half of the records exhibit a significant trend at local scale.

    (3) Changed to "multiple" tree-ring datasets.

**Comment 10:**

    (1) P9 L70-72 Instead of presenting the number of overlapping tree-ring chronologies it would be more helpful to report a percentage (although this might be more difficult under a constantly changing number of records).

    (2) "Although this might be more difficult under a constantly changing number of records". We agree and therefore we don't consider to report a percentage to be a useful information here.

    (3) No change.

**Point- by – Point response Anonymous**

**Comment 1:**

    (1) With regard to 1, the authors appropriately discuss the latitudinal gradient in the orbitally forced trends in temperature. They nevertheless do little to describe and investigate the latitudinal sampling biases in the two populations that they explore, namely the dendro and composite proxy populations. This sampling bias is obvious in Figure 1 and in the sample sizes listed in Figure 5. And yet figures like Figure 2 are presented with little caveat. Such a figure is misleading, given that the composite records are biased toward the high latitudes and the dendro records are biased toward the midlatitudes and the lower midlatitudes in particular (incidentally, it is not mentioned anywhere whether these means are themselves weighted by cos(lat), as they should be). How should we interpret these time series given that the explored effect intensifies at the higher latitudes? Splitting Figure 2 into time series representing different latitude bands would help (30 degree boxes may be too large for this), as would a scatter plot of trends vs. latitude for each of the two populations. While not definitive, it would be helpful for understanding how spatial sampling of a spatially-dependent temperature trend may be biasing the mean trends estimated from the two populations. I suspect that the authors will bring up Figure 5 as a rebuttal to what I am pointing out, but please see my comments on my second principle concern below.

(2) Figure 2 is merely a reproduction of a figure shown in the Pages publication (and in response with suggestions of reviewer 1, only the NH is shown). The figure is not part of our analysis, but demonstrates that we can reproduce the Pages trends. The latitudinal sampling bias is indeed tested Figure 5, but we added another splitting as suggested.

(3) Supplement figure added showing composite chronologies after splitting by latitude.

**Comment 2:**

(1) Before I get to that, however, I would add that another overlooked bias is that of the proxies comprising the composite records. They all sample different seasonal windows, some reflect marine temperature changes as opposed to continental changes, and many have their own biases tied to representation of low-frequencies. The authors take the composite proxies at face value, presumably because they fit their assumptions about latitudinal trends (in most cases), but it is insufficient to do so.

(2) This is obviously a critique on the data composition and/or papers using this by combining proxies, which is exactly what we try to make readers aware of. Here we point to systematic differences among proxies, and even include a splitting by season.

(3) No changes made.

**Comment 3**

There may be biases in these other records that promote spurious trend estimates that the authors do nothing to highlight. One observation that may point to such biases is the increase in the percentage of significant mid-latitude trends in the composite records relative to the high latitude records, which is of course counter to the expected spatial dependence. These factors are not sufficiently discussed. Regarding concern 2, the authors present Figure 5 as a measure of the latitudinal differences in the significance testing of trends in the dendro and composite records. The percentage of each population with significant and insignificant trends is nevertheless hard to interpret. Some additional significance testing would go a long way toward helping to interpret this figure and the results. The first question that should be addressed is: given the expected magnitude of trends estimated a priori from the orbitally-forced changes in insolation (signal), time series with the level of variance representative of the proxy series (noise), and the size of the sample populations, how many times would one achieve significant positive/negative trends and insignificant positive/negative trends for different realizations of noise? For instance, it may be the case that for 16 time series and the level of variance that is estimated in each, the trend percentages in the dendro high-latitude bin is actually what you would expect for a modestly detectable trend. Moreover, how should we interpret the comparison between the percentages associated with the dendro and composite series in a band like the high latitudes in Figure 5? It may be that the PDF of the percentage distributions spans the

differences shown in Figure 5 and the results are fully consistent with each other. Put differently, for multiple noise realizations, how robust is the separation between the trend percentages in the dendro and composite records? It is impossible to answer these questions from what the authors have done and they should better characterize the statistical likelihood that the differences they describe are more than just noise. All of the above is fundamental for two reasons. The first is that the difference in the number of statistically significant trends is the primary metric by which the authors argue there are differences in the representations of trends between the two populations. If the physical expression of those trends is latitudinally dependent and their spatial sampling is heterogenous and biased in the two populations, it must be controlled for. Secondly, the robustness of the differences must also be statistically constrained so that real differences (statistically speaking) can be separated from differences that can arise simply by chance.

> (3) We added a test and (supplementary) figure addressing the problem of latitudinal sample biases. We randomly (1000 times) drew 10 series from all tree ring series, and the composite (glacier, marine and lake records), at the latitudinal bands 0-90°N, 30-60°N and 60-90°N (below 30°N sample replication is too low), and calculated the percent of records showing insignificant/significant cooling and insignificant/significant warming trends.

**Minor comments:**

**Comment 4:**

> (1) Pg. 1, Ln. 13: It should be noted here that the 692 proxies are the temperature sensitive records in the database (the full database is closer to 3000).
> (2) No, the database includes exactly 692 records. We refer to PAGES 2k Consortium: A global multiproxy database for temperature reconstructions of the Common Era, Nat. Sci. Dat., 4, 1-33, 2017.
> (3) No changes.

**Comment 5:**

> (1) Pg. 1, Ln. 24: There are a lot more reviews that speak to this issue than Frank et al. Consider adding Jones et al. (2009), North et al. (2006), Mann (2007), Smerdon and Pollack (2016), and Christiansen and Ljungqvist (2017).
> (3) Further references added.

**Comment 6:**

(1) Pg. 2, Ln 28: The list of multiproxy reconstructions does not include the data assimilation work (e.g. Hakim et al. (2016) and Steiger et al. (2018)) nor does it include the PAGES products from 2013 and 2018. This should be corrected.

(3) Corrected.

**Comment 7:**

(1) Pg. 2, Ln 32: The list of review articles that discuss this should be expanded as above.

(3) Expanded.

**Comment 8:**

(1) Pg. 2, Ln 34: This is once again a limited list of papers that compare reconstructions and models. The authors should at least include the PAGES efforts from PAGES2k- PMIP Group (2015) and PAGES Hydro2K Consortium (2017), if not include some of the additional references that are discussed in those studies.

(3) 3 references added.

**Comment 9:**

(1) Pg. 4, Ln 6: Consider discriminating instead of critical

(3) Changed, accordingly.

**Comment 10:**

(1) Pg. 5, Ln 46: It seems strange to use cubic smoothing splines for standardization in the context of this investigation, given that they will explicitly remove the long-term trends. The effect is clearly visible in Figure 6 where even the 20th century trends have been removed. Incidentally, I find the bracket and description in Figure 6b a bit clumsy and hard to follow. The bracket in particular looks like it was drawn in by hand!

(3) Spline detrending was removed and replaced by Signal Free detrending in line with suggestions made by reviewer 1. Bracket replaced with an arrow.

**Comment 11:**

(1) Pg. 6, Ln. 84-86: Doesn't this contradict a central premise of the paper? This seems a lot more concerning than the attention it is given in the manuscript.

(2) No, this is "just" related to removing cambial/biological age-trends inherent to tree-ring width and density data.

(3) No changes.

**Comment 12:**

(1) Pg. 7, Ln 90: The subset is described as 70 but there are multiple places where this number appears to be different. Figure 5, for instance, discusses 89 dendro series. Are these typos or am I missing something?

(3) Further explanation added.

**Comment 13:**

(1) Pg. 7, Ln. 5: How does -0.32 compare to -0.03?!

(3) Removed.

**Comment 14:**

(1) Pg. 7, Ln. 9: Consider preserving instead of conserving

(3) Changed.

**Comment 15:**

(1) Pg. 8, Ln. 37: I find the discussion starting here and extending to the end of the paragraph very confusing. It seems to be saying that the authors have demonstrated differences between proxies, but that there are no differences between proxies. With regard to the last sentence, I do not think the authors have demonstrated the lack of spatial sampling bias, based on the principle arguments I have provided above.

(2) See above, response to comment #3.

**Point- by – Point response Freddy Ljungqvist (not for Clim. Past) – No official Review / Private suggestions.**

**Comment 1:**

(1) Page 2, Line 24: Would also, besides Frank et al., cite: https://doi.org/10.1002/wcc.418
Inserted.

**Comment 2:**

(1) Page 2, Lines 28–29: Would also add: https://doi.org/10.1126/science.1177303 And replace Christiansen and Ljungqvist (2011) with: https://doi.org/10.5194/cp-8-765-2012
Done.

**Comment 3:**

(1) Page 2, Line 34: Add https://doi.org/10.1175/JCLI-D-18-0525.1
Done.

**Comment 4:**

(1) Page 3, Lines 67–69: I'm not sure if it a good idea to just cite a list with references to different reconstructions with different amplitude of low-frequency temperature variability. In my opinion, it would be better to cite articles that discusses the mechanisms/reasons for these differences.
Not Done.

**Comment 5:**

(1) Page 3, about lines 76–79 and page 5, lines 32–35 (and other places): I do not agree that summer-temperature sensitive proxies necessary would capture orbital cooling trends the best. The forcing change is during summer but the climatic effect may be as strong in other seasons due to feedback mechanisms. See, for example, https://doi.org/10.5194/cp-6-609-2010 Actually, I would remove the discussions about seasonality altogether from the article as it would make a very complicated discussion about feedback mechanisms necessary to include it. Honestly, I think the expectation that orbital-induced cooling would be strongest in summer (only because the change in forcing occurred during these season) may be flawed – at least it is controversial. It exists conflicting evidence for which season that the orbitally

25  forced mid-Holocene Thermal Maximum made the warmest. Some lines of evidence suggests it was winter or annual mean temperature and not in summer for the high-latitudes in the NH (although much pollen data and model simulations mostly indicate it was summer). This said, orbital cooling still ought to be found in summer too of course.

30  **Comment 6:**

(1) Page 4, Line 97: I would insert "e.g." before Seim et al. (2012) as it is an example only.
Done.

**Comment 7:**

35  (1) Page 4, Line 102: I rather think a mixed temperature and hydroclimate sensitivity is a problem here. To this problem you may cite:

Babst F et al 2013 Site- and species-specific responses of forest growth to climate across the European continent. Global Ecol. Biogeogr. 22 706–17

Babst F et al 2019 Twentieth century redistribution in climatic drivers of global tree growth. Sci. Adv. 5
40  eaat4313

Galván JD, Camarero JJ, Ginzler C, Büntgen U 2014 Spatial diversity of recent trends in Mediterranean tree growth. Environ. Res. Lett. 9 084001

Klesse S et al 2018 A combined tree ring and vegetation model assessment of European forest growth sensitivity to interannual climate variability. Global Biogeochem. Cy. 32 1226–40

45  Done.

**Comment 8:**

(1) Page 7, line 14: At some place – but not here – I would recommend to make an even larger issue of the low r-values for Asia. The correlations are barely significant on average!
50  At some place – but not here… ?

**Comment 9:**

(1) Page 8, line 18: I would expand this discussion of different sample strategies. Happy to talk to you about it.
55  Not done.

**Comment 10:**

    (1) Page 8, line 33: Add citation: https://doi.org/10.1038/s41586-019-1060-3
       Added.

**Comment 11:**

    (1) Page 8, section 4.1. This section needs revision. The data–model mismatch could be due to a lack of certain feedback mechanisms in the models. Actually, the same "mismatch" problem exists for many regions outside the North Atlantic region (see, for example, https://doi.org/10.5194/cp-2018-145). Moreover, borehole data supports an even larger annual mean temperature increase during the mid-Holocene globally (1–1.5°C relative to pre-industrial). The borehole estimates are totally independent from other proxies and their noise sources. See: https://doi.org/10.1029/2008GL034187
       ?

**Comment 12:**

    (1) Page 9, line 54: Guess it should be late twentieth century and early twenty-first century temperatures.
       Added.

**Comment 13:**

    (1) Page 9, line 66: Mann et al. (1999) is NOT discussing this issue. Improper reference here
       Removed.

[revised manuscript text omitted]

**Fig. S.4.** Effects of orbital forcing on low-frequency temperature trends. Uncertainty estimates of a selection of plots displayed in Fig. 5. Randomly 1000 times, 10 **(a)** tree-ring and **(b)** marine, lake sediment and glacier ice records from the latitudinal bands 0-90N, 60-90N and 30-60N were selected. The fraction of 50-year binned records that exhibit a significant negative (dark blue) and non-significant cooling (blueish) trend or significant (red) and non-significant (reddish) warming trend at p < 0.05 over the pre-industrial (1-1800 CE) and derived from the statistical significance of the slope of least-squares linear regressions through each individual 50-year binned proxy record was assessed.

[Figure]

**Fig. S.5.** Compilation of NH and at least 800 year-long temperature-sensitive proxy records from the PAGES 2k initiative. 50-year binned composites from different latitudinal bands, 0-90°N (black), 30-60°N (green), and 60-90°N (blue) including **(a)** marine sediment, lake sediment and glacier ice records expressed in standard deviation units. Straight lines highlight the pre-industrial temperature trends (1-

 1800 CE) and lower panels show the corresponding temporal distribution of the records. Grey shadings indicate 95% bootstrap confidence intervals with 500 replicates. **(b)** same as in for tree-rings.

75 **Table S.1.** Information about 67 tree-ring records used for the detrending test, listed in and retrieved from Pages 2k 2017 (Pages 2k Consortium, 2017) metadata base.

| Series | Lat | Lon | Country | Site Name | Proxy | First | Last |
|--------|-----|-----|---------|-----------|-------|-------|------|
| Arc 008 | 67.90 | -140.70 | Canada | Yukon | TRW | 1177 | 2000 |
| Arc 061 | 66.90 | 65.60 | Russia | Polar Urals | MXD | 891 | 2006 |
| Arc 062 | 68.26 | 19.60 | Sweden | Tornetrask | MXD | 557 | 2008 |
| Arc 065 | 66.30 | 18.20 | Sweden | Arjeplog | ΔDensity | 1200 | 2010 |
| Arc 079 | 66.80 | 68.00 | Russia | Yamalia | TRW | 914 | 2003 |
| Asi 048 | 36.30 | 98.08 | China | CHIN006 | TRW | 159 | 1993 |
| Asi 049 | 37.00 | 98.08 | China | CHIN005 | TRW | 840 | 1993 |
| Asi 051 | 35.07 | 100.35 | China | MQAXJP | TRW | 1082 | 2001 |
| Asi 052 | 34.78 | 99.78 | China | MQBXJP | TRW | 470 | 2002 |
| Asi 053 | 34.72 | 99.67 | China | MQDXJP | TRW | 1163 | 2001 |
| Asi 077 | 38.70 | 99.68 | China | HYGJUP | TRW | 540 | 2006 |
| Asi 084 | 37.47 | 97.23 | China | CHIN050 | TRW | 843 | 2001 |
| Asi 085 | 37.47 | 97.22 | China | CHIN051 | TRW | 828 | 2001 |
| Asi 086 | 37.45 | 97.53 | China | CHIN052 | TRW | 404 | 2002 |
| Asi 087 | 37.43 | 98.05 | China | CHIN053 | TRW | 451 | 2002 |
| Asi 088 | 37.45 | 97.78 | China | CHIN054 | TRW | 711 | 2003 |
| Asi 094 | 37.32 | 98.40 | China | CHIN060 | TRW | 943 | 2003 |
| Asi 095 | 37.03 | 98.63 | China | CHIN061 | TRW | 857 | 2003 |
| Asi 096 | 37.03 | 98.67 | China | CHIN062 | TRW | 845 | 2001 |
| Asi 097 | 36.75 | 98.22 | China | CHIN063 | TRW | 681 | 2001 |
| Asi 098 | 36.68 | 98.42 | China | CHIN064 | TRW | 900 | 2001 |
| Asi 119 | 30.33 | 130.45 | Japan | JAPA018 | TRW | 1141 | 2005 |
| Asi 125 | 40.17 | 72.58 | Kyrgyzstan | KYRG007 | TRW | 1157 | 1995 |
| Asi 127 | 39.92 | 71.47 | Kyrgyzstan | KYRG009 | TRW | 1019 | 1995 |
| Asi 129 | 39.83 | 71,50 | Kyrgyzstan | KYRG011 | TRW | 694 | 1995 |
| Asi 145 | 48.35 | 107.47 | Mongolia | MONG021 | TRW | 996 | 2002 |
| Asi 175 | 27.78 | 87.27 | Nepal | NEPA030 | TRW | 856 | 1996 |

| | | | | | | | |
|---|---|---|---|---|---|---|---|
| Asi 195 | 36.33 | 74.03 | Pakistan | PAKI006 | TRW | 1032 | 1993 |
| Asi 196 | 36.33 | 74.03 | Pakistan | PAKI007 | TRW | 1141 | 1993 |
| Asi 202 | 36.58 | 75.08 | Pakistan | PAKI009 | TRW | 476 | 1990 |
| Asi 203 | 36.58 | 75.08 | Pakistan | PAKI010 | TRW | 968 | 1990 |
| Asi 204 | 36.58 | 75.08 | Pakistan | PAKI011 | TRW | 554 | 1990 |
| Asi 205 | 36.58 | 75.08 | Pakistan | PAKI012 | TRW | 1069 | 1990 |
| Asi 211 | 35.17 | 75.50 | Pakistan | PAKI015 | TRW | 736 | 1993 |
| Asi 212 | 35.17 | 75.50 | Pakistan | PAKI016 | TRW | 388 | 1993 |
| Asi 221 | 31.12 | 97.03 | China | CHIN046 | TRW | 449 | 2004 |
| Asi 222 | 29.07 | 93.95 | China | CHIN044 | TRW | 1047 | 1993 |
| Asi 224 | 30.30 | 91.52 | China | CHIN048 | TRW | 1080 | 1998 |
| Asi 227 | 24.53 | 121.38 | Taiwan | TW001 | TRW | 907 | 2007 |
| Asi 229 | 12.22 | 108.73 | Vietnam | VIET001 | TRW | 1030 | 2008 |
| Eur 003 | 68.00 | 25.00 | Finland | NSCAN | MXD | 1 | 2006 |
| Eur 004 | 49.00 | 20.00 | Slovakia | Tatra | TRW | 1040 | 2011 |
| Eur 007 | 46.40 | 7.80 | Switzerland | Lötschental | MXD | 755 | 2004 |
| Eur 008 | 44.00 | 7.50 | France | French Alps | TRW | 969 | 2007 |
| NAm 001 | 35.30 | -111.40 | USA | San Franciso Peaks | TRW | 1 | 2002 |
| NAm 002 | 67.10 | -159.60 | USA | Kobuk/Noatak | TRW | 978 | 1992 |
| NAm 003 | 60.50 | -148.30 | USA | Prince William Sound | TRW | 873 | 1991 |
| NAm 007 | 36.50 | -118.20 | USA | Flower Lake | TRW | 898 | 1987 |
| NAm 008 | 36.30 | -118.40 | USA | Timber Gap Upper | TRW | 699 | 1987 |
| NAm 009 | 36.30 | -118.20 | USA | Cirque Peak | TRW | 917 | 1987 |
| NAm 011 | 37.20 | -118.10 | USA | Sheep Mountain | TRW | 1 | 1990 |
| NAm 013 | 37.80 | -119.20 | USA | Yosemite National P. | TRW | 800 | 1996 |
| NAm 018 | 36.30 | -118.30 | USA | Boreal Plateau | TRW | 831 | 1992 |
| NAm 019 | 36.40 | -118.20 | USA | Upper Wright Lakes | TRW | 1 | 1992 |
| NAm 026 | 51.40 | -117.30 | Canada | Athabasca | MXD | 1072 | 1991 |
| NAm 029 | 52.70 | -118.30 | Canada | Bennington | TRW | 1104 | 1996 |
| NAm 030 | 50.80 | -115.30 | Canada | French Glacier | TRW | 1069 | 1993 |
| NAm 032 | 60.20 | -138.50 | Canada | Landslide | TRW | 913 | 2001 |
| NAm 044 | 45.30 | -111.30 | USA | Yellow Mountain Ridge | TRW | 470 | 1998 |
| NAm 045 | 46.30 | -113.20 | USA | Flint Creek Range | TRW | 999 | 1998 |
| NAm 046 | 46.00 | -113.40 | USA | Pintlers | TRW | 1200 | 2005 |
| NAm 049 | 40.20 | -115.50 | USA | Pearl Peak | TRW | 320 | 1985 |
| NAm 050 | 38.50 | -114.20 | USA | Mount Washington | TRW | 825 | 1983 |

| NAm 071 | 37.00 | -116.50 | USA | Great Basin Composite | TRW | 1 | 2009 |
| NAm 104 | 68.70 | -141.60 | USA | Firth River 1236 | MXD | 1073 | 2002 |
| NAm 151 | 52.20 | -117.20 | Canada | Athabasca Glacier 2 | TRW | 920 | 1987 |
| NAm 203 | 41.40 | -106.20 | USA | Sheep Trail | TRW | 1097 | 1999 |

---

## Author Response (AR2)

**Point-by-point Response**

**Reviewer 1**

The revised version of the Klippel et al. manuscript "Differing pre-industrial cooling trends between tree-rings and lower resolution temperature proxies" improved in many aspects. The authors also addressed many of my comments very comprehensively in the point-by-point response.

_–/–_

However, it is disappointing that even my main concerns (#2 and #3) did not result in any changes in the manuscript although the authors performed the suggested additional tests. I still think that this is important information for future readers of this article, because - binning+scaling is the more appropriate way of dealing with differences in resolution (of course I understand that the authors did scaling+binning instead to reproduce the PAGES figure). The differences in the figures are quite interesting and it would be good to have binning+scaling results as a supplementary figure.

- changing the bin size apparently affects the significance of trends quite strongly. It is fine to use the 50 years in the ms, but mentioning the tests and results with 200 years seems indicated, too.

The test of reversing the application of scaling and binning was added to the supplement (Fig. S4). A description why, we kept the original approach was inserted in the section material in methods. The test of using 200 year binned samples was also added to the supplement (Fig. S3).

Regarding comment 5, the authors are still using the term "inappropriate detrending" on P26, L71, which I find inappropriate for the reasons expressed in comment 5. Since the authors agreed on my arguments, it should be changed to "individual (series) detrending".

Changed accordingly.

Overall, many sections that were changed during revision require a thorough language correction. I found a couple of typos and grammar issues.

We checked the manuscript again for mistakes and made corrections.

**Reviewer 2**

The manuscript has improved from its original form, but the authors have not fully addressed my concerns. The issue of spatial sampling biases has not been fully addressed, while several additional issues arise. I also list a collection of minor issues that must be addressed before the paper can be published. My recommendation is to send it back to the authors for another major revision based on my comments below.

_–/–_

The sampling bias has simply not been addressed. The bootstrap sampling that the authors perform does not evaluate the differences in the spatial sampling of the two proxy populations. It doesn't matter how many subsamples the authors use if they are drawing from two populations that are inherently biased toward two different geographic distributions. It would be much more useful to simply have a scatter plot of the proxy trends as a function of latitude. The data should be color-coded for the two populations (dendro and other), thus allowing the reader to see the range and density of latitudinal sampling and whether there is a trend dependence on latitude evidenced in the two populations. Such a plot is necessary and should be included among the results that the authors present. As I indicate below, the percent of records with significant trends is also not a very useful statistic when considering populations with sampling biases. Binning the trends in 5-degree latitude bands and then calculating the trend in the trend vs. latitude data for the two populations may be much more useful.

In our revised manuscript, we now show the trend as a function of latitude through a new figure added to the Supplementary document (Fig. S6c). We also experimented with binning trends over more narrow 5-degree latitudinal bands, but unfortunately, the number of records changes substantially from one bin to another and even includes bands without any data (please see the below table). Because of this data limitation, we feel it is more meaningful and realistic to use broader bands (specifically 0-30°N, 30-60°N, 60-90°N).

Table 1: Number of NH records per 5° latitudinal bands from glacier ice, marine and lake sediment and tree ring records >800 years.

| Lat °N | n Tree | n Glacier/Marine/Lake |
|--------|--------|------------------------|
| 00-05 | 0 | 2 |
| 05-10 | 0 | 2 |
| 10-15 | 1 | 3 |
| 15-20 | 0 | 1 |
| 20-25 | 1 | 8 |
| 25-30 | 2 | 1 |
| 30-35 | 9 | 7 |
| 35-40 | 40 | 5 |
| 40-45 | 4 | 7 |
| 45-50 | 11 | 10 |
| 50-55 | 5 | 0 |
| 55-60 | 0 | 2 |
| 60-65 | 5 | 11 |

| | | |
|---|---|---|
| 65-70 | 10 | 14 |
| 70-75 | 1 | 4 |
| 75-80 | 0 | 8 |
| 80-85 | 0 | 4 |
| 85-90 | 0 | 0 |

A new issue that emerges after reading the paper again is that the authors equate the presence of trends with low-frequency variability. These two things are not equivalent and the authors should be careful in their discussion. Spectral analyses should be included if the authors want to make the kind of definitely statements that they do about the expression of low-frequencies in the two populations.

We agree with the point that linear trends overlap but are not exactly the same as „low-frequency variability". We now use the term "trend" only when referring to trend line derived from least-squares linear regression applied to individual (50-year binned) proxy records. The terms "low-frequency variability" and "long-term variability" we use only when describing fluctuations in the composite chronology. The latter is common praxis in late Holocene paleoclimatology.

I also am surprised that the interpretation of trends is tied entirely to orbital forcing. The LIA is not mentioned at all, and will have a control over trends, particularly those computed over the last 800-1000 years. The expression of the LIA was heterogeneous and therefore adds to the importance of spatial sampling in the two populations considered. The LIA should be mentioned and the authors should explain why they feel comfortable interpreting trends exclusively through the lens of orbital forcing and not with a consideration of the LIA. Incidentally, given the prevailing focus on volcanism as a cause of the LIA and the associated hemispheric and latitudinal expressions of the many eruptions during the LIA, this is really something that should be given consideration in the manuscript.

We actually do not just focus on orbital, but also tested potential influences due to detrending (chapter 3.2) and signal strength (chapter 3.3). However, since some of the series do not extend back to the putative MWP, the spatially varying LIA magnitude could influence the results, and this potential bias is now mentioned more clearly in the revision. Beyond this, the argument of spatial variability would also apply to the putative MWP, as well as the putative LALIA, and so on, i.e. a pronounced MWP or Roman Warm Period might enhance pre-industrial cooling trends. So, varying record length is a potential bias regardless of the underlying, short-term forcings including solar and/or volcanic. Another effect that might be even more important, and that is now more clearly emphasized in the revision, refers to sample replication. This is particularly the case in tree-ring records that were not truncated a minimum replication of $n > 5$ (or else), which typically inflates variance and affect low-frequency trends. In the current revision new information was added to the discussion (4.4 Remaining uncertainties) emphasizing how we clearly acknowledge that multiple factors can influence trends retained in tree-ring data..

A couple hold overs from the first review: (1) Pg. 2, Ln 28: The list of multiproxy reconstructions does not include the data assimilation work (e.g. Hakim et al. (2016) and Steiger et al. (2018)) nor does it include the PAGES products from 2013 and 2018. This should be corrected.

Added.

Pg. 2 Ln 33: defensible not defendable

Changed.

Ln 41: from: not from;

Changed.

Ln 43: no comma after tree/borehole (1)

Changed.

Ln 54: A flat trend is the absence of a trend. Flat trend is nonsensical.

Changed.

Pg. 3 Ln 65: Why would a globally averaged estimate of temperature be expected to have the same negative trend as a record from northern Scandinavia where the orbital forcing would be expected to be maximized. There is a stark contrast between two such records only in as much as there would indeed be a trend difference, but for entirely expected reasons. I don't think such a comparison and contrast should indicate anything significant about the fidelity of the underlying tree-ring records.

Changed to: "The lack of a long-term negative trend in the average global tree-ring record could be related to the difficulty of retaining such low-frequency variance in dendrochronological timeseries (Cook et al. 1995). Esper et al. (2012) demonstrated that orbital trends are retained in a long and well-replicated maximum latewood density (MXD) chronology, whereas such variability could not be preserved in the tree-ring width (TRW) data from the same trees."

Pg. 5 Ln 35. How is the normalization of records with different resolutions handled? This makes it sound like records were first normalized and then averaged in 50-yr bins. But a normalized record with annual resolution will effectively be in different standardized units than a normalized record with decadal resolution. This is concerning given the nature of the comparison the authors are making, given that the dendro records will all have annual resolution, while the other records will have lower resolutions. The authors should confirm that they first binned and then normalized and not the other way around.

We now did both, and added a figure to the supplement to show the effects (Fig. S4). We also kept the normalization-and-binning procedure, as this is the approach conducted in the Pages 2k 2017 paper, and deviating from this would seemingly constrain our unbiased comparison.

Ln 50: I do not understand the description here and it raises another concern with the analysis. I thought that all records of at least 800 years were selected and all trends were measured over the 800-yr period. But this sentence makes it sound like trend calculations were performed over variable time periods, i.e. some 800 years and others longer. If that is the case, the trends are hard to compare and may be influenced strongly by events prior to the 800-year window (especially when considering regression leverage) in a way that trends calculated only over 800 years are not. The authors need to clarify this and I would strongly argue for a common analysis window over which all of the trends should be calculated.

We included the suggestion (Fig.5b) and added one additional test using only those records that span the Common Era (Fig.5c)

Ln. 55: (in-) significant cooling/cooling as a formulation is impossible to follow. This sentence makes no sense to me.

Typographical error. Changed to (in-)significant cooling/warming.

Pg. 7 Ln 01: It is these kinds of statistics that are hard to parse when there is uneven spatial sampling. Imagine the extreme case in which all the dendro records were in the tropics (admittedly impossible) and all the other records were from the Arctic. In such a case 100% of the high latitude records should have significant trends while 0% of the dendro records should have trends. Reporting such statistics would say nothing about the fidelity of trends in the records and have a clear physical explanation. While the divisions are not as strong here, I am simply unclear on how to interpret these percentages when the sampling biases exist as they do.

Please see our earlier response to spatial variability in the proxy network. Regrettably, there are no millennial length dendro records in the tropics and this may be a situation that will never change.

Pg. 9 Lns 67-69: I do not think this statement about internal variability is supported by the authors' results. Are they really suggesting that internal variability explains the differences in the records they have analyzed?

No, we don't suggest that internal variability is the main culprit but rather state that both internal and external forcings are relevant: "
[revised manuscript text omitted]

[Figure]

75    Fig. S4. Same as in Fig.2 (upper panel). In the lower panel, the binning and normalization procedure were reversed: First glacier ice (blue), lake sediment (red), marine sediment (orange) and tree (ring) records were set to a 50-year resolution and in a second step records were normalized over their individual length.

80

[Figure]

Fig. S5. Effects of orbital forcing on low-frequency trends. Uncertainty estimates of a selection of plots displayed in Fig. 5a. Randomly 1000 times, 10 (a) tree-ring and (b) marine, lake sediment and glacier ice records from the latitudinal bands 0-90N, 60-90N and 30-60N were selected. The fraction of 50-year binned records that exhibit a significant negative (dark blue) and non-significant cooling (blueish) trend or significant (red) and non-significant (reddish) warming trend at $p < 0.05$ over the pre-industrial (1-1800 CE) and derived from the statistical significance of the slope of least-squares linear regressions through each individual 50-year binned proxy record was assessed.

85

[Figure]

**Fig. S6.** Compilation of NH and at least 800 year-long temperature-sensitive proxy records from the PAGES 2k initiative. 50-year binned composites from different latitudinal bands, 0-90°N (black), 30-60°N (green), and 60-90°N (blue) including (a) marine sediment, lake sediment and glacier ice records expressed in standard deviation units. Straight lines highlight the pre-industrial trends (1-1800 CE) and lower panels show the corresponding temporal distribution of the records. Grey shadings indicate 95% bootstrap confidence intervals with

500 replicates. (b) Same as in a for tree-rings. (c) Pre-industrial trend as a function of NH latitude. Black dots indicate marine sediment, lake sediment and glacier ice records and green dots are tree-ring records.

[Figure]

**Fig. S7.** Relationship between the slope over the pre-industrial period (1-1800 CE) and the absolute length of the tree-ring, glacier ice, marine and lake sediment records from the NH. Red refers to a significant warming, reddish to an non-significant warming, blueish to an non-significant cooling and blue to a significant cooling.

Table S.1. Information about 67 tree-ring records used for the detrending test, listed in and retrieved from Pages 2k 2017 (Pages 2k Consortium, 2017) metadata base.

| Series | Lat | Lon | Country | Site Name | Proxy | First | Last |
|---|---|---|---|---|---|---|---|
| Arc 008 | 67.90 | -140.70 | Canada | Yukon | TRW | 1177 | 2000 |
| Arc 061 | 66.90 | 65.60 | Russia | Polar Urals | MXD | 891 | 2006 |
| Arc 062 | 68.26 | 19.60 | Sweden | Tornetrask | MXD | 557 | 2008 |
| Arc 065 | 66.30 | 18.20 | Sweden | Arjeplog | ΔDensity | 1200 | 2010 |
| Arc 079 | 66.80 | 68.00 | Russia | Yamalia | TRW | 914 | 2003 |
| Asi 048 | 36.30 | 98.08 | China | CHIN006 | TRW | 159 | 1993 |
| Asi 049 | 37.00 | 98.08 | China | CHIN005 | TRW | 840 | 1993 |
| Asi 051 | 35.07 | 100.35 | China | MQAXJP | TRW | 1082 | 2001 |
| Asi 052 | 34.78 | 99.78 | China | MQBXJP | TRW | 470 | 2002 |
| Asi 053 | 34.72 | 99.67 | China | MQDXJP | TRW | 1163 | 2001 |
| Asi 077 | 38.70 | 99.68 | China | HYGJUP | TRW | 540 | 2006 |
| Asi 084 | 37.47 | 97.23 | China | CHIN050 | TRW | 843 | 2001 |
| Asi 085 | 37.47 | 97.22 | China | CHIN051 | TRW | 828 | 2001 |
| Asi 086 | 37.45 | 97.53 | China | CHIN052 | TRW | 404 | 2002 |
| Asi 087 | 37.43 | 98.05 | China | CHIN053 | TRW | 451 | 2002 |
| Asi 088 | 37.45 | 97.78 | China | CHIN054 | TRW | 711 | 2003 |
| Asi 094 | 37.32 | 98.40 | China | CHIN060 | TRW | 943 | 2003 |
| Asi 095 | 37.03 | 98.63 | China | CHIN061 | TRW | 857 | 2003 |
| Asi 096 | 37.03 | 98.67 | China | CHIN062 | TRW | 845 | 2001 |
| Asi 097 | 36.75 | 98.22 | China | CHIN063 | TRW | 681 | 2001 |
| Asi 098 | 36.68 | 98.42 | China | CHIN064 | TRW | 900 | 2001 |
| Asi 119 | 30.33 | 130.45 | Japan | JAPA018 | TRW | 1141 | 2005 |
| Asi 125 | 40.17 | 72.58 | Kyrgyzstan | KYRG007 | TRW | 1157 | 1995 |
| Asi 127 | 39.92 | 71.47 | Kyrgyzstan | KYRG009 | TRW | 1019 | 1995 |
| Asi 129 | 39.83 | 71,50 | Kyrgyzstan | KYRG011 | TRW | 694 | 1995 |
| Asi 145 | 48.35 | 107.47 | Mongolia | MONG021 | TRW | 996 | 2002 |
| Asi 175 | 27.78 | 87.27 | Nepal | NEPA030 | TRW | 856 | 1996 |
| Asi 195 | 36.33 | 74.03 | Pakistan | PAKI006 | TRW | 1032 | 1993 |
| Asi 196 | 36.33 | 74.03 | Pakistan | PAKI007 | TRW | 1141 | 1993 |
| Asi 202 | 36.58 | 75.08 | Pakistan | PAKI009 | TRW | 476 | 1990 |
| Asi 203 | 36.58 | 75.08 | Pakistan | PAKI010 | TRW | 968 | 1990 |

| | | | | | | | |
|---|---|---|---|---|---|---|---|
| Asi 204 | 36.58 | 75.08 | Pakistan | PAKI011 | TRW | 554 | 1990 |
| Asi 205 | 36.58 | 75.08 | Pakistan | PAKI012 | TRW | 1069 | 1990 |
| Asi 211 | 35.17 | 75.50 | Pakistan | PAKI015 | TRW | 736 | 1993 |
| Asi 212 | 35.17 | 75.50 | Pakistan | PAKI016 | TRW | 388 | 1993 |
| Asi 221 | 31.12 | 97.03 | China | CHIN046 | TRW | 449 | 2004 |
| Asi 222 | 29.07 | 93.95 | China | CHIN044 | TRW | 1047 | 1993 |
| Asi 224 | 30.30 | 91.52 | China | CHIN048 | TRW | 1080 | 1998 |
| Asi 227 | 24.53 | 121.38 | Taiwan | TW001 | TRW | 907 | 2007 |
| Asi 229 | 12.22 | 108.73 | Vietnam | VIET001 | TRW | 1030 | 2008 |
| Eur 003 | 68.00 | 25.00 | Finland | NSCAN | MXD | 1 | 2006 |
| Eur 004 | 49.00 | 20.00 | Slovakia | Tatra | TRW | 1040 | 2011 |
| Eur 007 | 46.40 | 7.80 | Switzerland | Lötschental | MXD | 755 | 2004 |
| Eur 008 | 44.00 | 7.50 | France | French Alps | TRW | 969 | 2007 |
| NAm 001 | 35.30 | -111.40 | USA | San Franciso Peaks | TRW | 1 | 2002 |
| NAm 002 | 67.10 | -159.60 | USA | Kobuk/Noatak | TRW | 978 | 1992 |
| NAm 003 | 60.50 | -148.30 | USA | Prince William Sound | TRW | 873 | 1991 |
| NAm 007 | 36.50 | -118.20 | USA | Flower Lake | TRW | 898 | 1987 |
| NAm 008 | 36.30 | -118.40 | USA | Timber Gap Upper | TRW | 699 | 1987 |
| NAm 009 | 36.30 | -118.20 | USA | Cirque Peak | TRW | 917 | 1987 |
| NAm 011 | 37.20 | -118.10 | USA | Sheep Mountain | TRW | 1 | 1990 |
| NAm 013 | 37.80 | -119.20 | USA | Yosemite National P. | TRW | 800 | 1996 |
| NAm 018 | 36.30 | -118.30 | USA | Boreal Plateau | TRW | 831 | 1992 |
| NAm 019 | 36.40 | -118.20 | USA | Upper Wright Lakes | TRW | 1 | 1992 |
| NAm 026 | 51.40 | -117.30 | Canada | Athabasca | MXD | 1072 | 1991 |
| NAm 029 | 52.70 | -118.30 | Canada | Bennington | TRW | 1104 | 1996 |
| NAm 030 | 50.80 | -115.30 | Canada | French Glacier | TRW | 1069 | 1993 |
| NAm 032 | 60.20 | -138.50 | Canada | Landslide | TRW | 913 | 2001 |
| NAm 044 | 45.30 | -111.30 | USA | Yellow Mountain Ridge | TRW | 470 | 1998 |
| NAm 045 | 46.30 | -113.20 | USA | Flint Creek Range | TRW | 999 | 1998 |
| NAm 046 | 46.00 | -113.40 | USA | Pintlers | TRW | 1200 | 2005 |
| NAm 049 | 40.20 | -115.50 | USA | Pearl Peak | TRW | 320 | 1985 |
| NAm 050 | 38.50 | -114.20 | USA | Mount Washington | TRW | 825 | 1983 |
| NAm 071 | 37.00 | -116.50 | USA | Great Basin Composite | TRW | 1 | 2009 |
| NAm 104 | 68.70 | -141.60 | USA | Firth River 1236 | MXD | 1073 | 2002 |
| NAm 151 | 52.20 | -117.20 | Canada | Athabasca Glacier 2 | TRW | 920 | 1987 |
| NAm 203 | 41.40 | -106.20 | USA | Sheep Trail | TRW | 1097 | 1999 |

---

## Author Response (AR3)

**Point-by-point response**

This iteration of the manuscript is marginally improved over the previous version and I will not stand in the way of its publication any longer. There are some interesting analyses and various points worth including in the public literature. I nevertheless believe that the case the authors make is much weaker than some of their statements throughout and in some cases contradictory. The abstract, in particular, does not seem to have been updated based on the changes in the manuscript. It reads much more definitive in terms of the alleged absence of long-term trends in the tree-ring chronologies. For instance, the abstract states that during the 1-1800 CE interval "tree-ring proxies lack any evidence of a significant pre-industrial cooling" while the discussion states that "our results show that millennial-scale trends in NH proxy records are consistent between tree-ring, glacier ice and marine and lake sediment records when considering the period 1200-1800 CE to calculate the slope of pre-industrial trends but inconsistent between tree-rings and other proxies over the entire Common Era." I find the second statement much more qualified and not at all consistent with the definitive statement in the abstract that focuses only on the results from the 1-1800 interval.

The two statements highlighted here by the reviewer are not contradictory. These comments apply to two very distinct time intervals. When we review trends in proxies over the 1200-1800 period we see very little or no difference between the tree rings and other proxies. However, if we expand the period of analysis to encompass the 1-1800 CE period, then there are major differences between the proxies, with tree rings being the sole proxy that does not show evidence of long-term cooling. This second point is one of the main findings of the article and therefore appropriate to mention in the abstract.

It is also worth noting that the new Figure S6c makes clear what I have been saying all along, which is that it is hard to discern a latitudinal dependence in the pre-industrial cooling in either of the subset populations (if there is a dependence, it seems to be a decrease in the cooling trends with latitude!), a fact that is hard to resolve if orbital forcing were truly the explanation for the trend. Moreover, it is hard to see how one might conclude that significant differences exist between the tree-ring population and the other proxies. It is simply a fundamental figure that does not lend a lot of support to the authors' arguments. But Figure 5 also makes clear that not only does the 1200-1800 period produce similar trends across the dataset, even the 1-1800 assessment produces results in the very small population numbers that are quite similar, barring significance tests.

It is important to recognize that what is striking are the trends seen in the means and the proxy-differences as opposed to the deviations of single series. If the reviewer believes that tree-ring chronologies are capable of preserving millennial-scale trends, and that their mean does not deviate from that of other proxies, there is very little we can do at this point to convince otherwise. We believe these differences are rather crucial information that needs to be considered in large-scale reconstructions.

With all of the above said, the authors simply need to couch more of their conclusions about trends in the various proxies in caveats qualifying the uncertain nature of their results. Their work opens up some interesting questions about the representations of long-term trends in the various proxy data types, but I would not bet the proverbial farm on the robustness

35  of the differences in the long-term trends that they have characterized. The manuscript should be revised with those qualifications in mind.

We estimate uncertainties and clearly name the limitations of these analyses/data e.g.,

P.10 Ln 289-291 – " In contrast to glacier ice and marine and lake sediment records, most of the very longest tree-ring records covering the entire pre-industrial Common Era 1-1800 CE do not exhibit a long-term cooling. The high-latitude tree-ring based

40  temperature reconstruction from Scandinavia remains the only record with a significant pre-industrial cooling (Esper et al., 2012)."

P.11 Ln 321-327 – "The degree of similarity between the NEG tree-ring chronology produced here and the corresponding PAGES2k version suggests that the current PAGES2k tree-ring collection is not the most ideal for studying millennial scale

45  trends. This is in large part due to the limitations of individual series detrending (Cook et al., 1995). Even with the application of RCS and RCS-SIG (Briffa et al., 1992; Esper et al., 2003, Melvin et al., 2014), the detection of a millennial-scale cooling trend is still elusive. These findings clearly demonstrate that the limited low-frequency variance in tree-ring chronologies is not solely an artefact of individual series detrending, previously identified as main explanation for the observed lack of long-term oscillations in large scale temperature reconstructions (Esper et al., 2002)."

50

P.11/12 Ln 341-344 – "Such weak temperature sensitivities amongst the tree-rings is likely related to confounding non-climatic (Johnson et al., 2010; Konter et al., 2015) or hydroclimatic (Ljungqvist et al., 2016) growth controls, or to the circumstance that some records are by nature less sensitive to summer temperature than others (St. George, 2014)."

55  P.12. Ln 357-366 – "This work examines the influence of orbital forcing, tree-ring detrending and climate signal strength on pre-industrial cooling in marine and lake sediment, glacier ice and tree-ring proxy archives. In tree-ring chronologies, sample size decreases back in time, lowering the chronology's signal-to-noise ratio and increasing variance (Frank et al., 2007). A small sample size could create apparent trends in the composite chronology that are not real. Regrettably, critical information about the sample replication for each tree-ring chronology is not completely provided by the PAGES 2k initiative (PAGES2k

60  Consortium, 2017) and thus we speculate records were truncated according to community-wide standards. Furthermore, the influence of climate epochs during the Common Era; the Roman Optimum (Büntgen et al., 2011); the Medieval Warm Period and Little Ice Age (Grove, 1990) on pre-industrial temperature trends has not yet been systematically explored. Further exercise potentially requires assessment of the relationship between timing and magnitude of climate epochs and overall temperature trends (Frank et al., 2010)."

65

P.13 Ln 372-382– „Our analysis has shown that the observed discrepancies in long-term trends do not arise from the latitudinal and seasonally varying imprints of orbital forcing or the limited temperature sensitivity. Despite application of the most suitable tree-ring detrending, one that can potentially support the preservation of low-frequency temperature trends at

millennial time scales, substantial long-term trend differences between proxies remain. We conclude that some, possibly many
of the tree-ring records in the PAGES2k database are artificially limited in their low-frequency variance at centennial and
longer time scales due to individual series detrending. This observation is supported by the fact that when a more low-frequency
conserving tree-ring detrending method is applied to a large subset of suitable records, new corroborating evidence for the
existence of the LALIA appears. Such nuances in the effect various tree-ring detrending methods have on low-frequency
variance conservation needs to be considered when combining proxies in large scale temperature reconstructions to avoid the
underrepresenting late Holocene cooling trends prior to post-industrial warming in hemispheric and global mean temperature
reconstructions."

P.22 Ln 672 – „Grey shadings indicate 95% bootstrap confidence intervals with 500 replicates.
P.25 Ln 696 – „Shadings indicate 95% bootstrap confidence intervals with 500 replicates"
P.25 Ln 706 – "Light grey shadings indicate 95% bootstrap confidence intervals with 500 replicates"

I also include a few small details in need of correction below:
Pg. 5, Ln 46-49: I cannot make sense of what binning approach the authors actually used here. Please be specific about what
the PAGES2k method was (they say something about it in an earlier paragraph, but the details are vague). It is also not clear
why they need to adhere to the PAGES2k method, given the concerns they raise in the previous sentence. One wonders why
and the argument given is not a strong one.
The explanation can be found in the previous paragraph: "To account for the varying temporal resolution among the proxies,
from sub-annual to multi-decadal scale, all normalized records were averaged and set to the same resolution consisting of 50-
year bins (e.g. 1901-1950; 1951-2000; Fig. 4). To test the influence of bin size on low-frequency variability, the normalized
proxy records were also degraded to the 200-year resolution (Fig. S3). Test results show that bin size has no influence on the
strength of the pre-industrial trend."

In addition to the following, new, clarification.
Still, Ln 48 changed from "…we used their normalization and binning approach. Changed to "…we followed their approach,
first normalizing each record and then binning…"

Pg. 6, Ln 56: Milankovitch theory seems more appropriate than cycles here.
Done.

Pg. 12, Ln 63-64: It is rather rich sauce to first call in question the PAGES2k database and to have cautioned against the use
of multiproxy reconstructions derived from that database and then to use the Neukom et al. (2012) paper, a study based entirely
on PAGES2k multiproxy reconstructions, to further an argument here. Rich sauce indeed.

This is not about questioning the PAGES2k database or any other compilation effort, but about the characteristics and potential limitations of proxies, and how to improve large scale reconstructions by considering individual proxy strengths and weaknesses. This whole 'rich sauce' chapter 4.4 was included upon the reviewer's request. However, we have removed the Neukom et al. (2019…not 2012) reference, because we want to emphasize that this article is not about any specific reconstruction, but about the usage of the underlying data.

Pg. 13, Ln 77: detrending. not ,
Done.

Pg. 13, Ln 79: effect not affect
Done.

[revised manuscript text omitted]

Fig. S.5. Effects of orbital forcing on low-frequency trends. Uncertainty estimates of a selection of plots displayed in Fig. 5a. Randomly 1000 times, 10 (a) tree-ring and (b) marine, lake sediment and glacier ice records from the latitudinal bands 0-90N, 60-90N and 30-60N were selected. The fraction of 50-year binned records that exhibit a significant negative (dark blue) and non-significant cooling (blueish) trend or significant (red) and non-significant (reddish) warming trend at p < 0.05 over the pre-industrial (1-1800 CE) and derived from the statistical significance of the slope of least-squares linear regressions through each individual 50-year binned proxy record was assessed.

[Figure]

Fig. S.6. Compilation of NH and at least 800 year-long temperature-sensitive proxy records from the PAGES 2k initiative. 50-year binned composites from different latitudinal bands, 0-90°N (black), 30-60°N (green), and 60-90°N (blue) including (a) marine sediment, lake sediment and glacier ice records expressed in standard deviation units. Straight lines highlight the pre-industrial trends (1-1800 CE) and lower panels show the corresponding temporal distribution of the records. Grey shadings indicate 95% bootstrap confidence intervals with 500 replicates. (b) Same as in a for tree-rings. (c) Pre-industrial trend as a function of NH latitude. Black dots indicate marine sediment, lake sediment and glacier ice records and green dots are tree-ring records.

[Figure]

Fig. S7. Relationship between the slope over the pre-industrial period (1-1800 CE) and the absolute length of the tree-ring, glacier ice, marine and lake sediment records from the NH. Red refers to a significant warming, reddish to an non-significant warming, blueish to an non-significant cooling and blue to a significant cooling.

Table S.1. Information about 67 tree-ring records used for the detrending test, listed in and retrieved from Pages 2k 2017 (Pages 2k Consortium, 2017) metadata base.

| Series | Lat | Lon | Country | Site Name | Proxy | First | Last |
|--------|-----|-----|---------|-----------|-------|-------|------|
| Arc 008 | 67.90 | -140.70 | Canada | Yukon | TRW | 1177 | 2000 |
| Arc 061 | 66.90 | 65.60 | Russia | Polar Urals | MXD | 891 | 2006 |
| Arc 062 | 68.26 | 19.60 | Sweden | Tornetrask | MXD | 557 | 2008 |
| Arc 065 | 66.30 | 18.20 | Sweden | Arjeplog | ΔDensity | 1200 | 2010 |
| Arc 079 | 66.80 | 68.00 | Russia | Yamalia | TRW | 914 | 2003 |
| Asi 048 | 36.30 | 98.08 | China | CHIN006 | TRW | 159 | 1993 |
| Asi 049 | 37.00 | 98.08 | China | CHIN005 | TRW | 840 | 1993 |
| Asi 051 | 35.07 | 100.35 | China | MQAXJP | TRW | 1082 | 2001 |
| Asi 052 | 34.78 | 99.78 | China | MQBXJP | TRW | 470 | 2002 |
| Asi 053 | 34.72 | 99.67 | China | MQDXJP | TRW | 1163 | 2001 |
| Asi 077 | 38.70 | 99.68 | China | HYGJUP | TRW | 540 | 2006 |
| Asi 084 | 37.47 | 97.23 | China | CHIN050 | TRW | 843 | 2001 |
| Asi 085 | 37.47 | 97.22 | China | CHIN051 | TRW | 828 | 2001 |
| Asi 086 | 37.45 | 97.53 | China | CHIN052 | TRW | 404 | 2002 |
| Asi 087 | 37.43 | 98.05 | China | CHIN053 | TRW | 451 | 2002 |
| Asi 088 | 37.45 | 97.78 | China | CHIN054 | TRW | 711 | 2003 |
| Asi 094 | 37.32 | 98.40 | China | CHIN060 | TRW | 943 | 2003 |
| Asi 095 | 37.03 | 98.63 | China | CHIN061 | TRW | 857 | 2003 |

| Asi 096 | 37.03 | 98.67 | China | CHIN062 | TRW | 845 | 2001 |
| Asi 097 | 36.75 | 98.22 | China | CHIN063 | TRW | 681 | 2001 |
| Asi 098 | 36.68 | 98.42 | China | CHIN064 | TRW | 900 | 2001 |
| Asi 119 | 30.33 | 130.45 | Japan | JAPA018 | TRW | 1141 | 2005 |
| Asi 125 | 40.17 | 72.58 | Kyrgyzstan | KYRG007 | TRW | 1157 | 1995 |
| Asi 127 | 39.92 | 71.47 | Kyrgyzstan | KYRG009 | TRW | 1019 | 1995 |
| Asi 129 | 39.83 | 71,50 | Kyrgyzstan | KYRG011 | TRW | 694 | 1995 |
| Asi 145 | 48.35 | 107.47 | Mongolia | MONG021 | TRW | 996 | 2002 |
| Asi 175 | 27.78 | 87.27 | Nepal | NEPA030 | TRW | 856 | 1996 |
| Asi 195 | 36.33 | 74.03 | Pakistan | PAKI006 | TRW | 1032 | 1993 |
| Asi 196 | 36.33 | 74.03 | Pakistan | PAKI007 | TRW | 1141 | 1993 |
| Asi 202 | 36.58 | 75.08 | Pakistan | PAKI009 | TRW | 476 | 1990 |
| Asi 203 | 36.58 | 75.08 | Pakistan | PAKI010 | TRW | 968 | 1990 |
| Asi 204 | 36.58 | 75.08 | Pakistan | PAKI011 | TRW | 554 | 1990 |
| Asi 205 | 36.58 | 75.08 | Pakistan | PAKI012 | TRW | 1069 | 1990 |
| Asi 211 | 35.17 | 75.50 | Pakistan | PAKI015 | TRW | 736 | 1993 |
| Asi 212 | 35.17 | 75.50 | Pakistan | PAKI016 | TRW | 388 | 1993 |
| Asi 221 | 31.12 | 97.03 | China | CHIN046 | TRW | 449 | 2004 |
| Asi 222 | 29.07 | 93.95 | China | CHIN044 | TRW | 1047 | 1993 |
| Asi 224 | 30.30 | 91.52 | China | CHIN048 | TRW | 1080 | 1998 |
| Asi 227 | 24.53 | 121.38 | Taiwan | TW001 | TRW | 907 | 2007 |
| Asi 229 | 12.22 | 108.73 | Vietnam | VIET001 | TRW | 1030 | 2008 |
| Eur 003 | 68.00 | 25.00 | Finland | NSCAN | MXD | 1 | 2006 |
| Eur 004 | 49.00 | 20.00 | Slovakia | Tatra | TRW | 1040 | 2011 |
| Eur 007 | 46.40 | 7.80 | Switzerland | Lötschental | MXD | 755 | 2004 |
| Eur 008 | 44.00 | 7.50 | France | French Alps | TRW | 969 | 2007 |
| NAm 001 | 35.30 | -111.40 | USA | San Franciso Peaks | TRW | 1 | 2002 |
| NAm 002 | 67.10 | -159.60 | USA | Kobuk/Noatak | TRW | 978 | 1992 |
| NAm 003 | 60.50 | -148.30 | USA | Prince William Sound | TRW | 873 | 1991 |
| NAm 007 | 36.50 | -118.20 | USA | Flower Lake | TRW | 898 | 1987 |
| NAm 008 | 36.30 | -118.40 | USA | Timber Gap Upper | TRW | 699 | 1987 |
| NAm 009 | 36.30 | -118.20 | USA | Cirque Peak | TRW | 917 | 1987 |
| NAm 011 | 37.20 | -118.10 | USA | Sheep Mountain | TRW | 1 | 1990 |
| NAm 013 | 37.80 | -119.20 | USA | Yosemite National P. | TRW | 800 | 1996 |
| NAm 018 | 36.30 | -118.30 | USA | Boreal Plateau | TRW | 831 | 1992 |
| NAm 019 | 36.40 | -118.20 | USA | Upper Wright Lakes | TRW | 1 | 1992 |

| NAm 026 | 51.40 | -117.30 | Canada | Athabasca | MXD | 1072 | 1991 |
|---------|-------|---------|--------|-----------|-----|------|------|
| NAm 029 | 52.70 | -118.30 | Canada | Bennington | TRW | 1104 | 1996 |
| NAm 030 | 50.80 | -115.30 | Canada | French Glacier | TRW | 1069 | 1993 |
| NAm 032 | 60.20 | -138.50 | Canada | Landslide | TRW | 913 | 2001 |
| NAm 044 | 45.30 | -111.30 | USA | Yellow Mountain Ridge | TRW | 470 | 1998 |
| NAm 045 | 46.30 | -113.20 | USA | Flint Creek Range | TRW | 999 | 1998 |
| NAm 046 | 46.00 | -113.40 | USA | Pintlers | TRW | 1200 | 2005 |
| NAm 049 | 40.20 | -115.50 | USA | Pearl Peak | TRW | 320 | 1985 |
| NAm 050 | 38.50 | -114.20 | USA | Mount Washington | TRW | 825 | 1983 |
| NAm 071 | 37.00 | -116.50 | USA | Great Basin Composite | TRW | 1 | 2009 |
| NAm 104 | 68.70 | -141.60 | USA | Firth River 1236 | MXD | 1073 | 2002 |
| NAm 151 | 52.20 | -117.20 | Canada | Athabasca Glacier 2 | TRW | 920 | 1987 |
| NAm 203 | 41.40 | -106.20 | USA | Sheep Trail | TRW | 1097 | 1999 |